# CatSperζ regulates the structural continuity of sperm Ca²⁺ signaling domains and is required for normal fertility

Jean-Ju Chung[1,2,3]*, Kiyoshi Miki[1], Doory Kim[4,5], Sang-Hee Shim[4,5†], Huanan F Shi[3], Jae Yeon Hwang[3], Xinjiang Cai[6], Yusuf Iseri[1], Xiaowei Zhuang[4,5], David E Clapham[1,2]*

[1]Howard Hughes Medical Institute, Boston Children's Hospital, Boston, United States; [2]Department of Neurobiology, Harvard Medical School, Boston, United States; [3]Department of Cellular and Molecular Physiology, Yale School of Medicine, New Haven, United States; [4]Howard Hughes Medical Institute, Department of Chemistry and Chemical Biology, Harvard University, Cambridge, United States; [5]Department of Physics, Harvard University, Cambridge, United States; [6]Department of Medicine, James J. Perters VA Bronx, Icahn School of Medicine at Mount Sinai, New York, United States

*For correspondence: jean-ju. chung@yale.edu (J-JC); dclapham@enders.tch.harvard. edu (DEC)

Present address: [†]Department of Chemistry, Korea University, Seoul, Republic of Korea

Competing interests: The authors declare that no competing interests exist.

**Abstract** We report that the *Gm7068* (*CatSpere*) and *Tex40* (*CatSperz*) genes encode novel subunits of a 9-subunit CatSper ion channel complex. Targeted disruption of *CatSperz* reduces CatSper current and sperm rheotactic efficiency in mice, resulting in severe male subfertility. Normally distributed in linear quadrilateral nanodomains along the flagellum, the complex lacking CatSperζ is disrupted at ~0.8 μm intervals along the flagellum. This disruption renders the proximal flagellum inflexible and alters the 3D flagellar envelope, thus preventing sperm from reorienting against fluid flow *in vitro* and efficiently migrating *in vivo*. Ejaculated *CatSperz*-null sperm cells retrieved from the mated female uterus partially rescue *in vitro* fertilization (IVF) that failed with epididymal spermatozoa alone. Human CatSperε is quadrilaterally arranged along the flagella, similar to the CatSper complex in mouse sperm. We speculate that the newly identified CatSperζ subunit is a late evolutionary adaptation to maximize fertilization inside the mammalian female reproductive tract.

## Introduction

Sperm hyperactivation, characterized by a large asymmetric lateral displacement of the flagellum (*Ishijima et al., 2002*), is required for normal mammalian sperm navigation (*Demott and Suarez, 1992*), rheotaxis (*Miki and Clapham, 2013*), and *zona pellucida* (ZP) penetration (*Stauss et al., 1995*). Calcium influx through the flagellar Ca²⁺ ion channel, CatSper, triggers hyperactivation (*Carlson et al., 2003*; *Kirichok et al., 2006*; *Ren et al., 2001*) and leads to changes in the flagellar envelope during capacitation (*Chung et al., 2011*; *Quill et al., 2003*). In hyperactivated spermatozoa, the transverse flagellar force is larger than the propulsive flagellar force due to the increase in mid-piece curvature (α angle), which enables a larger range of motion and typical figure-of-eight swimming trajectories compared to the nearly straight paths of non-hyperactivated spermatozoa (*Ishijima, 2011*). Transverse force facilitates sperm penetration through the cumulus and ZP

**eLife digest** Male mammals ejaculate millions of sperm cells each time they mate with a female. Only a few of these cells manage to travel up the female's reproductive tract to reach the egg, and usually only one sperm fertilizes it. Freshly ejaculated sperm are incapable of fertilizing eggs and have to undergo several changes within the female to become able to do so. One crucial change occurs in the sperm tail, which starts to beat vigorously in a whip-like motion. This type of movement – known as hyperactivated motility – enables the sperm to swim towards the egg, push through a sticky coating that surrounds it, and then burrow into it.

Hyperactivated motility is triggered when calcium ions enter the sperm cell via a specific channel protein known as CatSper, which is found in the membrane that surrounds the cell. CatSper channels form groups (known as complexes) with several other proteins that are arranged in a unique pattern of four straight 'stripes' running down the tail of the sperm. This arrangement is necessary for hyperactivated motility and mutations in the genes that encode these proteins can lead to infertility in males. The CatSper channel complex is known to contain seven proteins: four that form a pore through which calcium ions can enter, and three accessory proteins whose roles in hyperactivated motility are less clear.

Chung *et al.* identified two genes in mice that encode new accessory proteins in the CatSper channel complex named CatSper epsilon and CatSper zeta. Further experiments analyzed the role of CatSper zeta in more detail. Mutant males that lack CatSper zeta have fragmented patterns of CatSper stripes in the tails of their sperm. Moreover, fewer calcium ions were able to pass through the channels to enter the cell. Together, this made the sperm tail more rigid, which prevented it from moving efficiently within the female, resulting in reduced fertility. Chung *et al.* also found that the mutant sperm were less able to penetrate the egg than normal sperm.

During evolution, the gene that encodes CatSper zeta appeared first in mammals and may represent an adaptation that improved the chances of a sperm fertilizing the egg inside the reproductive tract of female mammals. Future challenges will be to explore how the CatSper channel assembles on the membrane of sperm and find out exactly how calcium ions trigger hyperactivated motility.

(*Ishijima, 2011*; *Yanagimachi, 1966*). Spermatozoa from all *CatSper*-null (*1–4* or *d*) males have smaller α angles than wild-type (wt spermatozoa upon capacitation (*Chung et al., 2011*; *Qi et al., 2007*). Consistently, *CatSper*-null mutant spermatozoa migrate inefficiently *in vivo* (*Chung et al., 2014*; *Ho et al., 2009*) and fail to penetrate the ZP (*Ren et al., 2001*).

Sperm rheotax against Fallopian tubular and isthmus fluid flow (*Miki and Clapham, 2013*). Rheotactic turning to reorient to directional flow depends on flagellar rolling, not the sperm head or its geometry, as demonstrated by the rheotaxis of headless mouse sperm (*Miki and Clapham, 2013*). CatSper channels form unique $Ca^{2+}$ signaling domains in linearly quadrilateral arrays along the principal piece of sperm flagella. The integrity of these domains is necessary to time and/or maintain hyperactivated motility (*Chung et al., 2014*). Thus, *CatSper1*-null sperm cannot rheotax due to defects in rolling (*Miki and Clapham, 2013*), and presumably exert less lateral force in escaping from epithelial walls (*Ho et al., 2009*) or in pushing cumulus cells aside. In general, however, there is a lack of understanding of the steps between CatSper-mediated calcium entry, $Ca^{2+}$-modified phosphorylation cascades, and the resulting structural changes underlying orchestrated flagellar movement.

Here, we reveal that the murine *Gm7068* (*C1orf101-like*) and *Tex40* genes encode two new subunits of the CatSper ion channel complex, CatSper epsilon (ε) and zeta (ζ), respectively. In this study, we focus primarily on CatSperζ's function. Genetic disruption of mammalian-specific CatSperζ reduces the CatSper current in the sperm flagellum and hyperactivated motility, resulting in severe subfertility. We use high speed video microscopy and digital image analysis to determine swimming trajectory and the flagellar waveform in detail. Surprisingly, abrogation of CatSperζ renders the proximal flagellum inflexible but preserves overall motility, thus resulting in restriction of the 3D flagellar envelope, inefficient sperm rheotaxis *in vitro*, and delayed sperm migration *in vivo*. Using

super-resolution microscopy, we demonstrated that the structurally distinct CatSper Ca$^{2+}$ signaling domains along the flagellum (*Chung et al., 2014*) becomes fragmented in the absence of *CatSperz*. We demonstrate that IVF failure of *CatSperz*-null spermatozoa is partially rescued by using ejaculated sperm recovered from the uterus of mated females, explaining the discrepancy between *in vitro* and *in vivo* fertilizing ability. Finally, we show that mouse and human spermatozoa have a similar macroscopic organization of the CatSper complex.

## Results

### CatSper ε and ζ: Two new accessory proteins in the CatSper channel complex

We previously identified seven protein components of the CatSper channel complex (CatSper1-4, *β*, *γ*, and δ) from mouse testis using tandem affinity purification (*Chung et al., 2011*). As the most biochemically complex ion channel known to date, it has not been possible to express functional CatSper channels in heterologous systems. This includes many attempts in many cell types, including simultaneous injection of all 7 *CatSper* mRNAs into *Xenopus* oocytes (*data not shown*). Therefore, we continued to seek potential additional components to more thoroughly understand CatSper channel assembly and trafficking. We identified a mouse homolog of human *C1orf101* (*C1orf101-like*, currently *Gm7068*) (*Figure 1A*) based on its sequence homology to the C-terminal extracellular domain of CatSperδ (*Figure 1—figure supplement 1A*). This testis-specific gene (*Figure 1—figure supplement 2A*) is predicted to encode a single transmembrane (TM) protein (*Figure 1C* and *Figure 1—figure supplement 1B,C*). In addition, a small soluble protein encoded by another testis-specific gene, *Tex40* (*Figure 1—figure supplement 2A*), was found to be associated with the CatSper channel complex (*Figure 1B and C*, and *Figure 1—figure supplement 1D*). In this study, we refer to the *C1orf101-like* and *Tex40* genes as *CatSpere* and *CatSperz*, respectively (see Molecular Cloning, Materials and methods). Like the other CatSper accessory subunits (*Chung et al., 2011*), both *CatSpere* and *CatSperz* mRNAs express specifically in germ cells and are detected before *CatSper1* expression during postnatal development (*Figure 1—figure supplement 2B,C*). Moreover, mouse CatSper ε and ζ proteins partition into the testis microsome fraction (P) (*Figure 1—figure supplement 2D*), complex with CatSper1, and exhibit interdependence with the expression of the other CatSper subunits (*Figure 1D–1F*). In both human and mouse sperm cells, CatSper ε and ζ proteins are localized to the principal piece of the tails (*Figure 1G and H* and *Figure 1—figure supplement 2E–G*).

### CatSper ε and ζ localize at quadrilateral Ca$^{2+}$signaling domains in sperm flagella

Mouse CatSper proteins form a unique pattern of four linear ('racing stripes') Ca$^{2+}$ signaling domains running down the four quadrants of the principal piece of the flagellum (*Chung et al., 2014*). We examined whether ε and ζ share this distinctive compartmentalization. The antibodies, anti-hε31, recognizing the N-terminal extracellular region of human CatSperε, and anti-mζ174, against the very C-terminus of mouse CatSperζ, were suitable for 3D stochastic optical reconstruction microscopy (STORM) (*Figure 1F–1H* and *Figure 1—figure supplement 2E*). CatSperζ and CatSperε show the apparent four-fold arrangement of CatSper1, *β* and δ subunits in mouse (*Figure 1I*) and human (*Figure 1J*) spermatozoa.

### *CatSperz*-null male mice have severely impaired fertility

The lack of functional expression of CatSper channels in heterologous systems requires that genetic manipulation be used to study the function of each component. *CatSpere* has the same ancient origin at the root of early eukaryotic evolution as those of *CatSpers1-4*, *b*, and *g* and the same pattern of extensive lineage-specific gene loss as *CatSperb* and *g* through metazoan evolution (*Figure 2—figure supplement 1A*) (*Cai et al., 2014*). While CatSper δ and ε share high C-terminal sequence homology (*Figure 1—figure supplement 1A*), CatSperδ appears later in evolution (*Figure 2—figure supplement 1A*). In contrast, CatSperζ has no conserved domains and, like hyperactivated motility, is only present in mammals (*Figure 2—figure supplement 1A*), leading us to speculate that CatSperζ is a required evolutionary adaptation to mammalian fertilization. Based on sequence

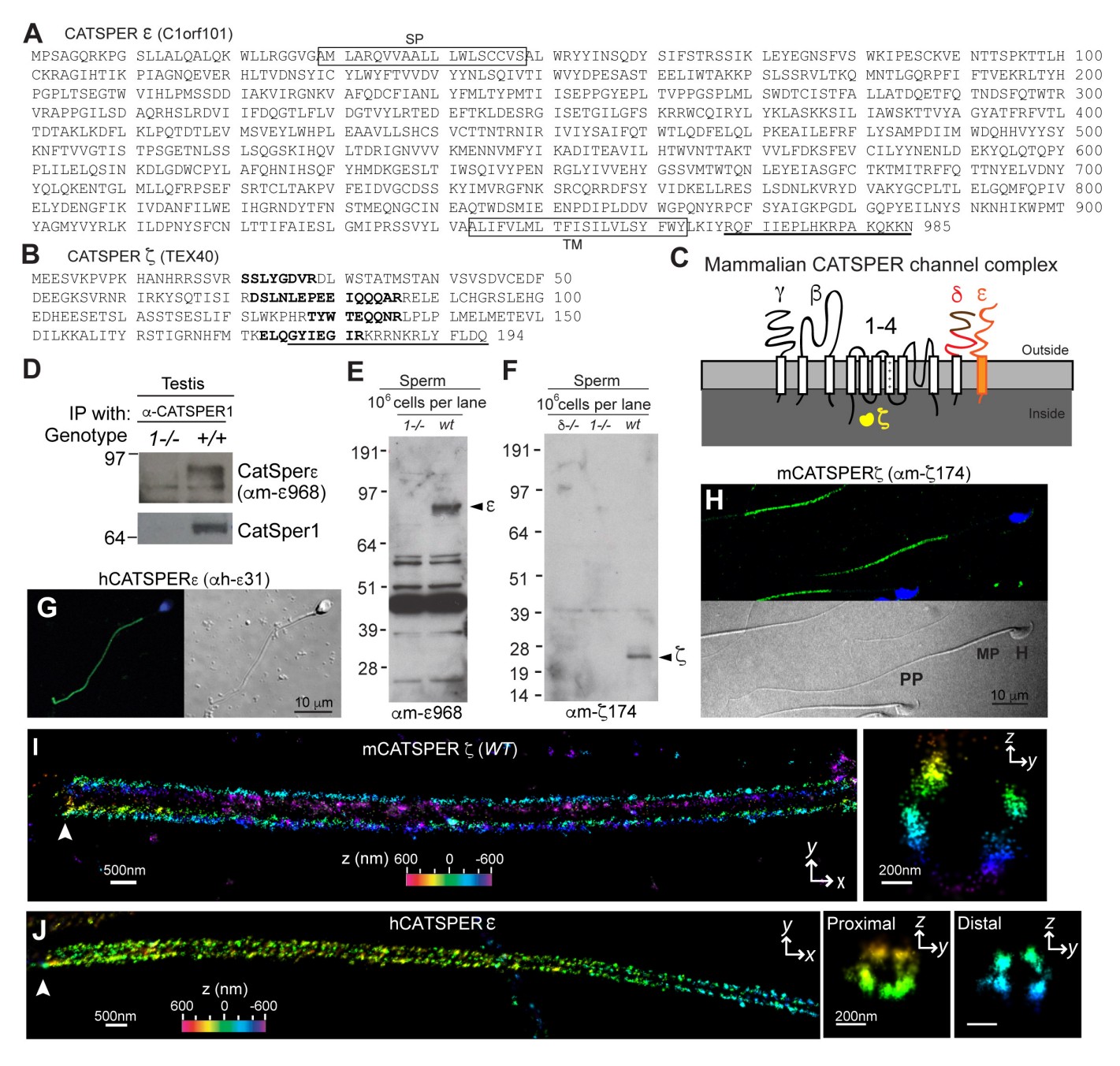

**Figure 1.** CatSper ε and ζ, two new accessory proteins of CatSper channel complex. (**A** and **B**) Mouse protein sequences of CatSper ε (**A**) and ζ (**B**). (**C**) Cartoon of the predicted topology of 9 CatSper subunits. (**D**) Association of CatSperε with CatSper1 in testis. (**E** and **F**) Dependence of CatSper ε (**E**) and ζ (**F**) proteins on CatSper1 in mouse sperm cells. (**G** and **H**) Confocal fluorescence and the corresponding phase-contrast images of immunostained human CatSperε (**G**) and mouse CatSperζ (**H**). (**I**) 3D STORM images of mouse CatSperζ in capacitated wt sperm. *x-y* projection (left) and a *y-z* cross-section (right) at 0.5 um from the annulus. The color encodes the relative distance from the focal plane along the *z* axis (color scale bar in *x-y* projection). (**J**) 3D STORM images of human CatSperε in *x-y* projection (left), in *y-z* cross-sections (right). Colors indicate the *z* positions (see color scale bar). See also *Figure 1—figure supplements 1–2*.

The following source data and figure supplements are available for figure 1:

**Figure supplement 1.** Identification of CatSper ε and ζ, two novel accessory proteins of the CatSper channel complex, related to *Figure 1*.

**Figure supplement 2.** Expression of *CatSper e* and *z* mRNAs and proteins, related to *Figure 1*.

*Figure 1 continued on next page*

*Figure 1 continued*

**Figure supplement 2—source data 1.** Temporal expression of *CatSper1*, *CatSper ε*, and *CatSper z* mRNAs during postnatal testis development.

homology and conservation, we anticipated that deletion of CatSperε would likely be the same as the existing knockout of other CatSper subunits, but deletion of CatSperζ might provide new insights into spermatozoan adaptations to changes concomitant with the evolution of mammalian fertilization. To test this idea, we began by generating a *CatSperz*-null mouse line from *Tex40* gene targeted ES cell clones. *Tex40* is a small gene composed of 5 exons that spans only ~3 kb on chromosome 11 (*Figure 2—figure supplement 1B*). Deletion of exons 2–4 was confirmed in the homozygous null mouse (*Figure 2—figure supplement 1C*). No CatSperζ protein was detected in *Tex40*-null spermatozoa by immunoblotting and immunocytochemistry (*Figure 2—figure supplement 1D, E*).

*CatSperz*-null mutant mice are indistinguishable from their wt or heterozygous (het) littermates in appearance, gross behavior, or survival. In addition, no morphological differences were observed by histological examination of testis and epididymis (*data not shown*). Sperm morphology and epididymal sperm number from *CatSperz*-null mice were not significantly different from those of 2–3 month old paired heterozygous littermates (*Figure 2—figure supplement 1E* and *Figure 2—figure supplement 2A*). *CatSperz*-null female mice exhibited normal mating behavior and gave birth to litters comparable to those of het females when mated with wt or het males (*Figure 2—figure supplement 2B*). However, when *CatSperz*-null male mice were mated with wt or het females, they were severely subfertile: 20% (5/25) *CatSperz*-null males were completely infertile over six months (*Figure 2A*), and progeny of the fertile paternal *CatSperz*-null mice were significantly fewer in number (*Figure 2B* and *Figure 2—figure supplement 2B*). The latency from pair formation to the birth of these offspring from *CatSperz*-null males was ≥10 days compared to those from wt or het males (*data not shown*).

We examined the number of sperm within cumulus oocyte complexes (COCs) after copulation and checked *in vivo* fertilization rates by isolating the COCs and/or embryos from the female ampullae. At 8 hr after coitus, no sperm was found in the COCs when mated with *CatSperz*-null male mice (*Figure 2C*). In contrast, the majority of the COCs from *CatSperz*-het mated females had one or more sperm cells within the complex. When mated with *CatSperz*-het males, more 2 cell eggs were observed over time after coitus, while the fertilization rate by *CatSperz*-null males did not change significantly (*Figure 2D* and *Figure 2—figure supplement 2C*). These data suggest that *CatSperz*-null sperm migration is delayed in the female reproductive tract.

## *CatSperz*-null sperm cells have rigid proximal flagella

To understand why *CatSperz*-null spermatozoa did not efficiently migrate in the female reproductive tract, we first investigated sperm motility using computer assisted sperm analysis (CASA) (*Figure 2—figure supplement 2D*). The percentage of motile spermatozoa was not significantly different and most motility parameters of *z*-null spermatozoa were comparable to those of *z*-het sperm cells. However, the characteristic increase of lateral head displacement upon capacitation was not observed in *z*-null spermatozoa (*Figure 2—figure supplement 2D*), suggesting that hyperactivated motility was reduced. Ninety minutes after capacitation, there was a less marked difference in swimming trajectories of *z*-null spermatozoa compared to *z*-het spermatozoa, supporting this notion (*Figure 2E* and *Video 1*). Further analysis of flagellar amplitude and waveforms of tethered spermatozoa revealed a striking rigidity of *z*-null spermatozoa from their midpiece to midway down the principal piece (*Figure 2F* and *Video 2*). This phenotype was also observed from hyperactivation-deficient *CatSper2*-null patients (*Smith et al., 2013*). After incubation under capacitating conditions for 90 min, we observed that *z*-null spermatozoa beat only at the very distal end of a flagellum (*Video 3*). Moreover, *CatSperz*-null spermatozoa remain bent in the anti-hook direction (*Ishijima et al., 2002*) (*Figure 2F*, *Videos 2* and *3*). The anti-hook bend predominates as the pro-hook bend (initiated by the CatSper-

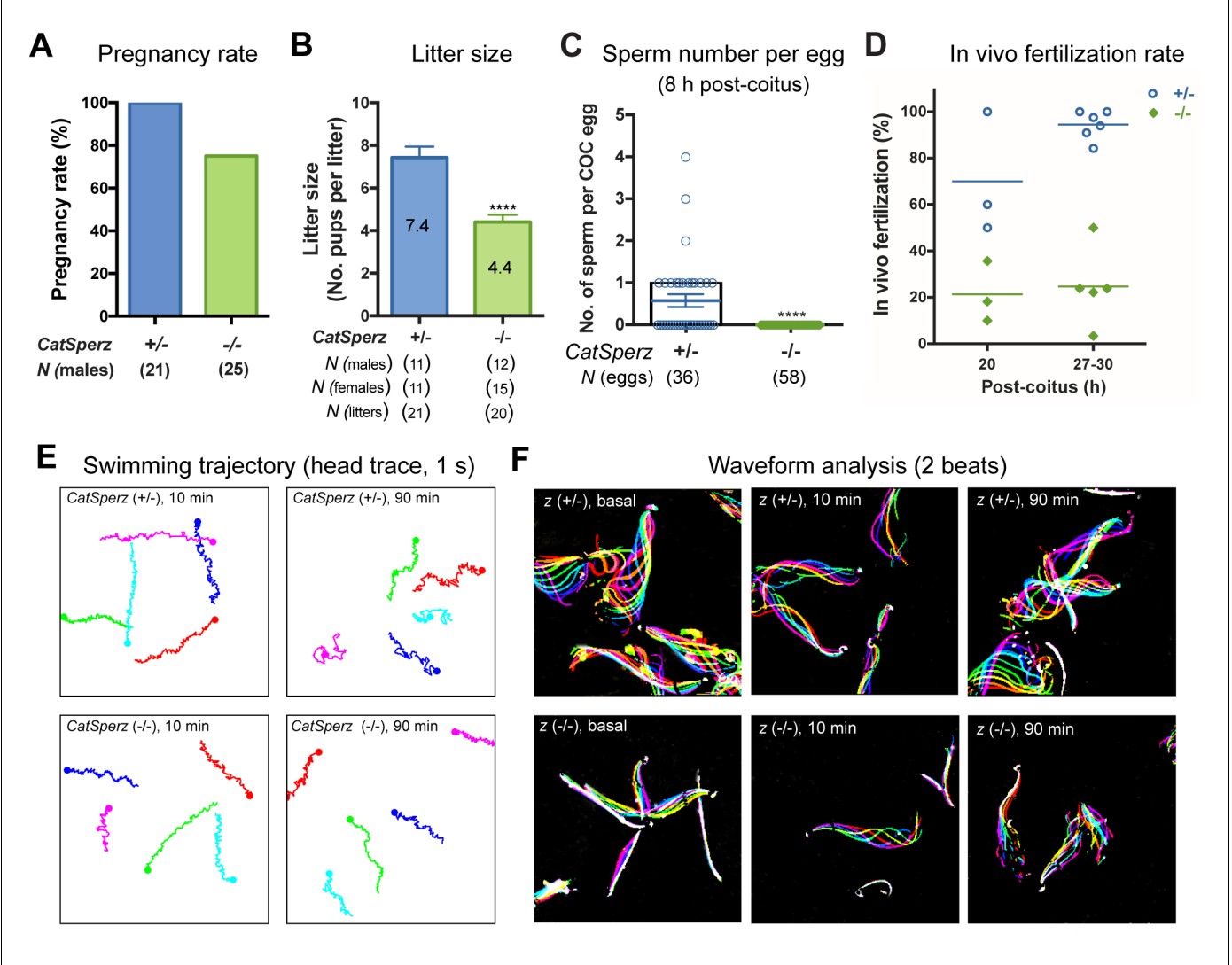

**Figure 2.** Deletion of the mouse CatSperζ subunit severely impairs male fertility. (**A**) Percent pregnancy rate over three months. (**B**) Average litter size resulting from *CatSperz*+/- (7.4 ± 0.5) and *CatSperz*-/- (4.4 ± 0.3) males. (**C**) Sperm number per egg at the fertilization site 8 hr after 1 hr window-timed coitus with *CatSperz*+/- (0.58 ± 0.15) and *CatSperz*-/- (0, none) males, quantified from eggs collected from ampullae. (**B**) and (**C**) Data are mean ± SEM. ****p<0.0001. (**D**) *In vivo* fertilization rate: Scatter plot with mean % of 2 cell fertilized eggs from *CatSperz*+/- (70% and 94.4%) and *CatSperz*-/- (21.3% and 24.6%) mated females at 20 and 27–30 hr after coitus, respectively. (**E**) Head trace of free swimming *CatSperz*+/- (top) and *CatSperz*-/- (bottom) sperm cells at 10 min (left) and 90 min (right) after capacitation. Traces are from 1 s movies taken at 37°C. (**F**) Flagellar waveform traces. Movies recorded at 200 fps: *CatSperz*+/- (top) and *CatSperz*-/- (bottom) sperm cells attached on glass coverslips before capacitation (left), and 10 min (middle), and 90 min (right) after capacitation. Overlays of flagellar traces from two beat cycles are generated by hyperstacking binary images; time coded in color. See also *Figure 2—figure supplements 1–2* and *Figure 6—figure supplement 1*.

The following source data and figure supplements are available for figure 2:

**Source data 1.** Impaired male fertility in *CatSperz*-/- mice: pregnancy rate, litter size, sperm number per egg, and *in vivo* fertilization rate.

**Figure supplement 1.** Generation of *CatSperz*-/- mice, related to *Figure 2*.

**Figure supplement 2.** Sperm count and fertility of *CatSperz*-/- mice; sperm motility analysis and development of P-Tyr, related to *Figures 2* and *3*.

**Figure supplement 2—source data 1.** Impaired male fertility in *CatSperz*-/- mice: sperm count, litter size per genotype, *in vivo* fertilization rate, and CASA parameters.

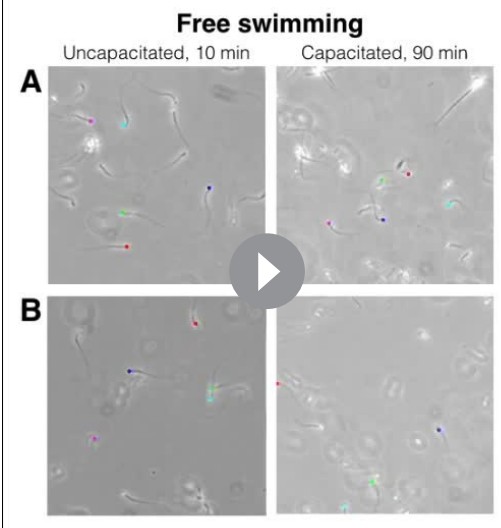

**Free swimming**

Uncapacitated, 10 min Capacitated, 90 min

**Video 1.** Movement of free swimming *CatSperz +/-* and *-/-* spermatozoa; related to *Figure 2*. Uncapacitated (left) and 90 min capacitated (right) spermatozoa were allowed to disperse for 10 min pre-incubation in a 37°C chamber containing HEPES-HTF; free swimming sperm cells recorded within the next 5 min; video rate 20 fps (1/5 speed), 1 s movies; head trace to track swimming trajectory. (A) *CatSperz+/-* and (B) *CatSperz-/-* spermatozoa.

mediated $Ca^{2+}$ signaling pathway (*Chang and Suarez, 2011*) is dysregulated in *CatSperz*-null spermatozoa.

## Reduced CatSper current in *CatSperz*-null spermatozoa

To examine how $Ca^{2+}$ signaling in *CatSperz*-null spermatozoa is impaired, we first examined $I_{CatSper}$, the sperm-specific $Ca^{2+}$-selective ion current. Since $Ca^{2+}$ has high affinity to calcium-selective pores (*Almers et al., 1984*), CatSper permeation of monovalents increases when external calcium is removed (*Kirichok et al., 2006; Navarro et al., 2007*). In divalent-free (DVF) solutions, wt spermatozoan $I_{CatSper}$ conducts a large $Na^+$ current, which is completely absent in mice lacking *CatSpers1, 2, 3, 4,* or *d* (*Chung et al., 2011; Kirichok et al., 2006; Qi et al., 2007*). However, in *CatSperz*-null spermatozoa, monovalent CatSper current is present but reduced to ~60% of normal ($-426 \pm 50$ pA at $-100$ mV; *Figure 3A*) compared to control *CatSperz*-het spermatozoa ($-683 \pm 77$ pA at $-100$ mV; *Figure 3B*). Thus, in the absence of CatSperζ, the CatSper channel complex is still targeted to the flagellar membrane and forms functional channels. We hypothesize that the reduction in CatSper current reflected decreased protein expression levels.

P2X receptors are nonselective ion channels gated by purines such as ATP. The ATP-activated cation-nonselective current in the midpiece of murine sperm is mediated by the $P_2X_2$ purinergic receptor (*Navarro et al., 2011*). In *CatSperz*-null spermatozoa, $I_{ATP}$ did not differ substantially from heterozygous spermatozoa (*Figure 3A and B*), supporting the assumption that there is selective down regulation of CatSper channels. Smaller $I_{CatSper}$ explains, in part, the attenuated hyperactivated motility, delayed sperm migration, and male sub-fertility (*Figure 2A–2E*). Protein tyrosine phosphorylation (P-Tyr), a hallmark of sperm capacitation, is

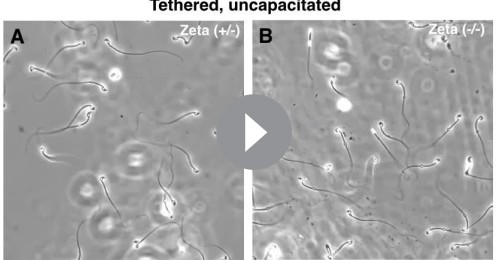

**Tethered, uncapacitated**

**Video 2.** Motility of tethered *CatSperz +/-* and *-/-* spermatozoa; uncapacitated, related to *Figure 2*. Uncapacitated epididymal spermatozoa in non-capacitating M2 media were tethered to the fibronectin-coated glass bottom dish; sperm motility was recorded at 37°C; video rate 200 fps, 2 s movies. (A) *CatSperz+/-* and (B) *CatSperz-/-* spermatozoa.

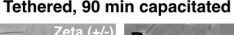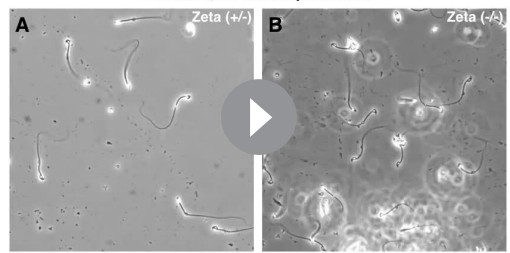

**Tethered, 90 min capacitated**

**Video 3.** Motility of tethered *CatSperz +/-* and *-/-* spermatozoa; 90 min capacitated, related to *Figure 2*. After 90 min incubation in HTF, capacitated epididymal spermatozoa were tethered to a fibronectin-coated glass bottom dish; sperm motility was recorded at 37°C; video rate 100 fps (1/2 speed), 1 s movies. (A) *CatSperz+/-* and (B) *CatSperz-/-* spermatozoa.

potentiated and delocalized in *CatSper* knockout mice (*Chung et al., 2014*) or when $Ca^{2+}$ influx is pharmacologically blocked (*Navarrete et al., 2015*). Upon capacitation, P-Tyr was more prominent in *CatSperz*-null spermatozoa than wt, but to a lesser extent than *CatSper1*-null spermatozoa (*Figure 2—figure supplement 2E*), consistent with the reduced calcium current. It is, however, also possible that an altered arrangement of the CatSper complex and/or its interaction with target proteins in the linear domains could have contributed to these functional deficits.

## Abrogation of *CatSperz* retards targeting of the CatSper complex to flagella

To better understand why $I_{CatSper}$ is reduced in *CatSperz-null* spermatozoa, we examined levels of protein expression in *CatSperz*-null spermatozoa (*Figure 4A*). Expression of other CatSper subunits was detected in *CatSperz* -null spermatozoa, albeit at 30–60% lower levels than that of wt (*Figure 4B*), consistent with reduced $I_{CatSper}$ (*Figure 3*). This contrasts with the complete absence of other CatSper subunits in *CatSper1*- and *CatSperd*-null spermatozoa. mRNA and protein levels of other CatSper subunits were not reduced in the testis of *CatSperz*-null mice (*Figure 4C and D*), suggesting that the defect occurs during or after assembly of the protein complex.

## *CatSperz* is essential in maintaining the continuity of linear flagellar CatSper Ca²⁺domains

Loss of *CatSperz* resulted in fragmentation of CatSper1 staining on the flagellar membrane and these defects are large enough to be resolved by confocal imaging (*Figure 5A*). These gaps were not observed in wt/het (*Figure 1H* and *Figure 2—figure supplement 1E*) or previous wt and CatSper knockout studies (*Chung et al., 2011*; *2014*; *Liu et al., 2007*; *Ren et al., 2001*). 3D STED and 3D STORM super-resolution microscopies clearly demonstrate that structural continuity is interrupted in *CatSperz*-null spermatozoa - each 'stripe' of the CatSper domains is fragmented, while the overall quadrilateral structure is maintained (*Figure 5B and C*). Cross-sections of the 3D STORM image of wt flagellum show the normal four tight clusters (*Figure 5C*, lower), represented as four lines in the 2D angular profiles of surface localizations (*Figure 5—figure supplement 1A,E*; inset) as previously observed (*Chung et al., 2014*). In *CatSperz*-null spermatozoa, however, the four lines in the 2D angular profiles were interrupted (*Figure 5—figure supplement 1B*). To examine whether the interruptions were periodic, we performed autocorrelation analysis and Fourier transform of STORM images of *CatSperz*-null sperm flagella (*Figure 5—figure supplement 1C–F*). Autocorrelation analysis of the *CatSperz*-null sperm flagella exhibited enhanced periodicity compared to the wt flagellum, with the first peak at ~850 nm (*Figure 5—figure supplement 1C,D*). The Fourier transform shows a fundamental frequency of $(800\ nm)^{-1}$ (*Figure 5—figure supplement 1F*). We assume this thinning of one or more linear domains reflects an underlying structural periodicity that regulates CatSper complex trafficking or membrane insertion.

## *CatSperz*-null spermatozoa rheotax inefficiently with reduced torque

Thus far, our results show that *CatSperz*-null sperm have reduced $I_{CatSper}$, dysregulated structural continuity of the CatSper $Ca^{2+}$ signaling domains, beat in an atypical pattern, and are delayed in migrating in the female reproductive tract, resulting in reduced male fertility. Rheotactic guidance for sperm over long distances requires rotational motion during CatSper-mediated hyperactivated motility (*Miki and Clapham, 2013*). We thus measured rheotactic parameters and the rotation rate of *CatSperz*-null spermatozoa with a particular focus on whether their proximal tail rigidity and subsequent low amplitude lateral movement (*Figure 2*) affects sperm movement. In flow-directed capillary tubes (*Miki and Clapham, 2013*), we observed that the rheotactic ability of *CatSperz*-null spermatozoa was significantly reduced (*Figure 6A and B* and *Figure 6—figure supplement 1A*). At all flow rates tested, most motile *CatSperz*-null spermatozoa were unable reorient to swim against the flow and were swept out of the tube (*Video 4*). In contrast, 85% of motile heterozygous spermatozoa displayed rheotactic behaviors by maintaining their position or swimming upstream for more than 2 s of the 9 s period of recording (*Figure 6B* and *Video 4*).

We next examined the rotational motion of *CatSperz*-null spermatozoa. At high viscosities (0.3% methyl cellulose (MC), cP = 6.7), uncapacitated *z*-het spermatozoa swim in circles (*Figure 6C*, left and *Video 5*), while capacitated *z*-het spermatozoa swim in a more linear path as they rotate around

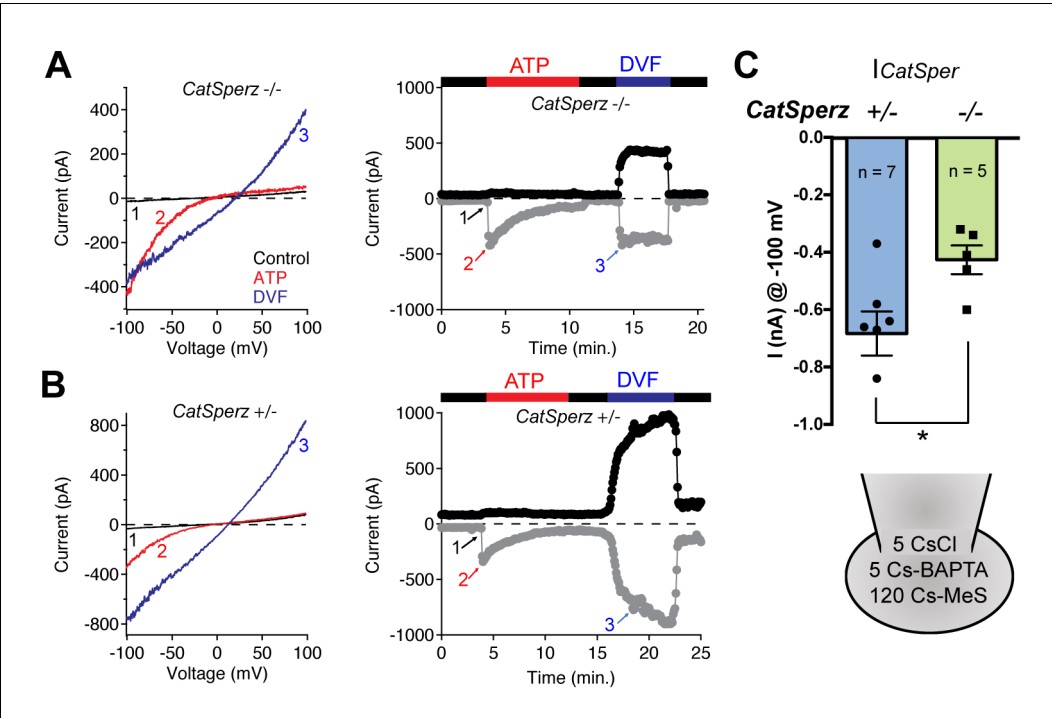

**Figure 3.** $I_{CatSper}$, but not ATP-activated P2X2 current, is reduced in *CatSperz*-null spermatozoa. (**A**) *CatSperz-/-* and (**B**) *CatSperz+/-* $I_{CatSper}$. Left panels show the current-voltage relations of monovalent $I_{CatSper}$ in response to voltage ramps at the time points indicated. Right traces are representative time courses of $I_{CatSper}$ measured in the standard bath solution (1, HS), ATP-activated P2X2 current (2, ATP), and nominally divalent-free solution (3, DVF) at −100 mV (gray circles) and +100 mV (black circles). $I_{CatSper}$ in *CatSperz*-null sperm cells is ~60% of that recorded from wt. Inward $I_{ATP}$ current induced by 100 μM ATP is similar in both phenotypes and indistinguishable from previously published wt $I_{ATP}$ (**Navarro et al., 2011**). (**C**) Average $I_{CatSper}$ measured from *CatSperz+/-* (−683 ± 77 pA) and *CatSperz-/-* (−426 ± 50 pA) sperm cells at −100 mV. Data are mean ± SEM. p=0.0297. Cartoon shows the standard pipette solution (mM); internal Cs used to block K$^+$ currents.

The following source data is available for figure 3:

**Source data 1.** Inward CatSper current at −100 mV.

a longitudinal axis, like wt spermatozoa (*Video 6*) (*Miki and Clapham, 2013*). Interestingly, *z*-null spermatozoa exhibit linear migration as they can rotate along the tail axis regardless of capacitating conditions, even at higher viscosities (*Figure 6C*, right and *Videos 5* and *6*). Indeed, uncapacitated *CatSperz*-null spermatozoa rotate ~50% faster than *z*-het spermatozoa (*Figure 6D*). This indicates that *CatSperz*-null spermatozoa have less lateral motion and are subject to less torque by the moving stream. Spatially, the spermatozoa trace out a less conical 3D envelope (*Figure 2F* and *Figure 6—figure supplement 1B*). In short, the rigidity of the *CatSperz*-null sperm proximal tail constrains its motion to that of a propeller-driven rod.

## Compromised Ca$^{2+}$signaling alters the sperm's 3D flagellar envelope and movement

We next examined the relation between external calcium entry and sperm function. First, we tested whether increasing extracellular [Ca$^{2+}$] could rescue *z*-null sperm motility. After incubation for 90 min with a two-fold greater [Ca$^{2+}$], most *CatSperz*-null sperm remain bent in the anti-hook orientation with a rigid proximal tail (*Figure 6—figure supplement 1C*, z(-/-) middle, and *Video 7*). A few *z*-null spermatozoa partially recover, bending occasionally in the pro-hook direction with hyperactivated motility (*Figure 6—figure supplement 1C,* z(-/-) right, and *Video 7*). Conversely, a 20-fold reduction

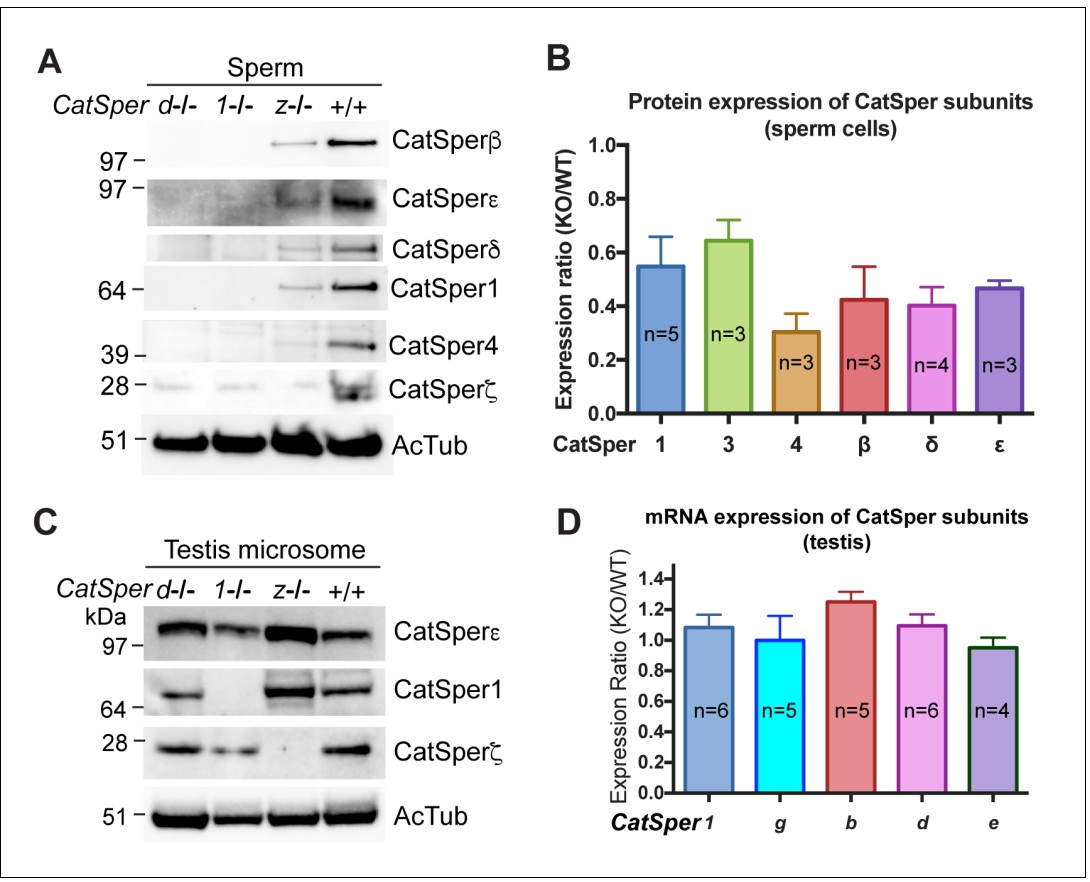

**Figure 4.** CatSper proteins are reduced in sperm from *CatSperz*-null mice despite protein expression during spermatogenesis. (**A** and **B**) Reduced expression of CatSper subunits in sperm cells of *CatSperz* homozygous null mice compared with their complete absence in *CatSper1* and *d*-null mice. Immunoblotting of (**A**) total mouse sperm extracts and (**B**) protein expression ratio (*z*-KO/wt) of CatSper 1 (0.5 ± 0.1), 3 (0.6 ± 0.08), 4 (0.3 ± 0.07), *β* (0.4 ± 0.1), *δ* (0.4 ± 0.07), and *ε* (0.5 ± 0.03). Data are mean ± SEM. (**C**) Increased expression of CatSper1 and ε in mouse testis in *CatSperz*-null mutants. (**D**) Quantitative gene expression analysis (qRT-PCR) from adult *CatSperz*-het and null testes: expression ratio ($2^{-ddCT}$) and mean ddCt (null-het); TATA binding protein (TBP) is the internal control. The expression ratio of *CatSper1* (1.1 ± 0.1) and all accessory *g* (1.0 ± 0.2), *b* (1.3 ± 0.07), *d* (1.1 ± 0.08), and *e* (0.95 ± 0.07) subunits are mean ± SEM.

The following source data is available for figure 4:

**Source data 1.** Protein and mRNA expression of CatSper subunits.

of extracellular [$Ca^{2+}$] alone did not significantly alter the flagellar waveforms of *z*-het spermatozoa within 90 min (*Figure 6—figure supplement 1C* and *Video 8*).

A transient $Ca^{2+}$ pulse induced by $Ca^{2+}$ ionophore, A23187, significantly reduces the time required for wt sperm to develop hyperactivated motility (*Tateno et al., 2013*). Moreover, a short (10 min) exposure to A23187 can rescue defects in hyperactivated motility and the fertilizing capability of *CatSper1*-null sperm *in vitro* (*Navarrete et al., 2016*). We tested the relation of calcium transients to hyperactivated motility in *CatSperz*-het and null sperm. In *z*-het spermatozoa, an A23187-induced $Ca^{2+}$ pulse followed by washout, enables full hyperactivation, characterized by wide lateral displacement with large midpiece *α* angle within 30 min (*Figure 6—figure supplement 1D* and *Videos 9* and *10*). However, in *z*-null sperm, the same treatment improved the flexibility of the proximal flagella, particularly in the principal piece, but the midpiece remained largely inflexible (*Figure 6—figure supplement 1D* and *Videos 9* and *10*). Building on our previous work, the present study suggests that calcium entry through CatSper channels has time-dependent, complex effects on the

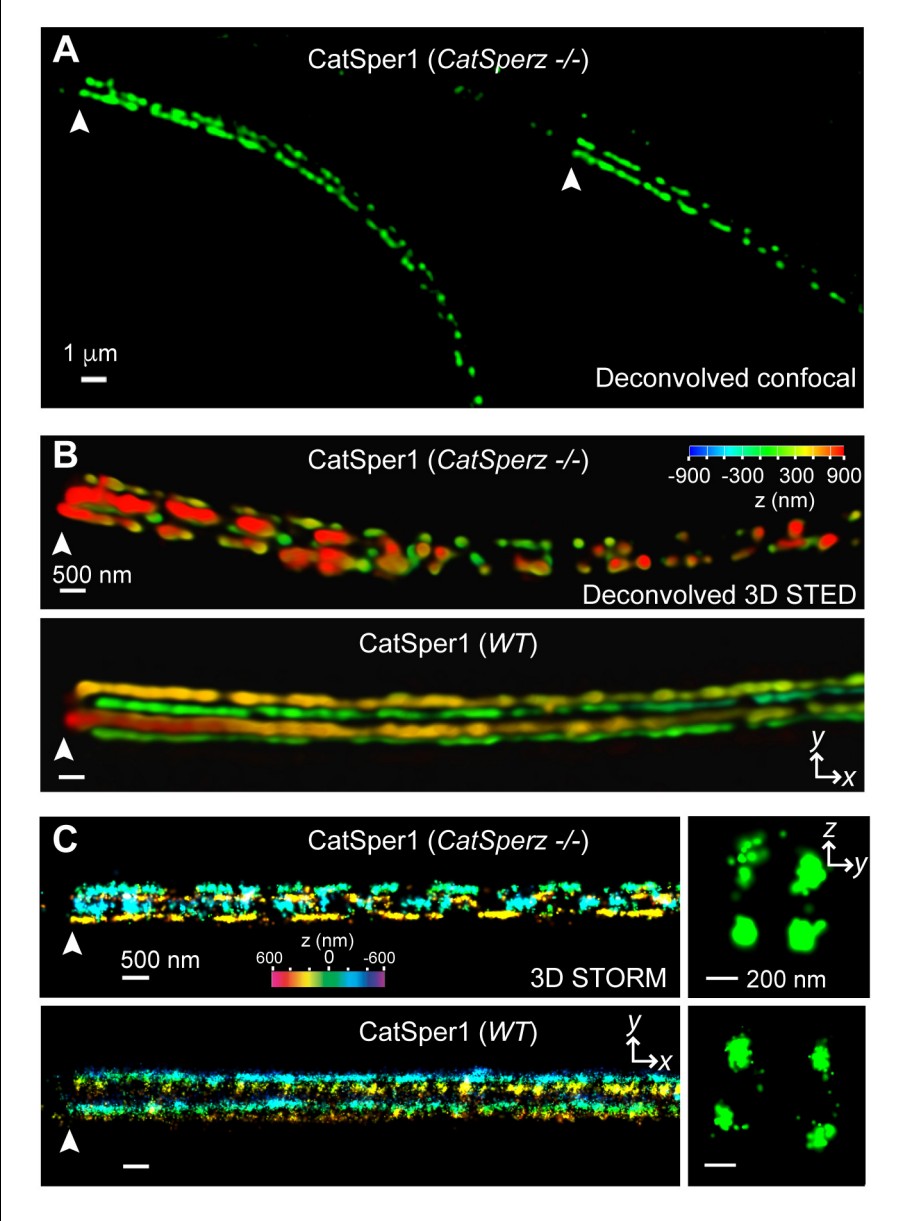

**Figure 5.** *CatSperz* deletion disrupts the continuity of the CatSper linear domains. Application of different modes of fluorescence microscopy to observe CatSper localization. (**A**) Deconvolved confocal image of α-CatSper1 immunostained *CatSperz* -null spermatozoa. Scale bar, 1 µm. (**B** and **C**) 3D super-resolution images of CatSper1. 3D STED (**B**) and 3D STORM (**C**) images of *CatSperz* -null (top) and wt (bottom) sperm flagella, respectively. *x-y* projection colors encode the relative distance from the focal plane along the *z* axis. Scale bar, 500 nm. Arrowheads indicate the junction between the mid-piece and the principal piece (annulus) of the tail. 3D STORM, *y-z* cross-section images are shown on the right. Scale bar, 200 nm. See also *Figure 5—figure supplement 1*.

The following figure supplement is available for figure 5:

**Figure supplement 1.** Subcellular distribution of immunolocalized CatSper proteins; related to *Figure 5*.

coordination of motility and that loss of *CatSperz* results in reduced $I_{CatSper}$, changes in calcium signaling, and structural alterations of the flagellum.

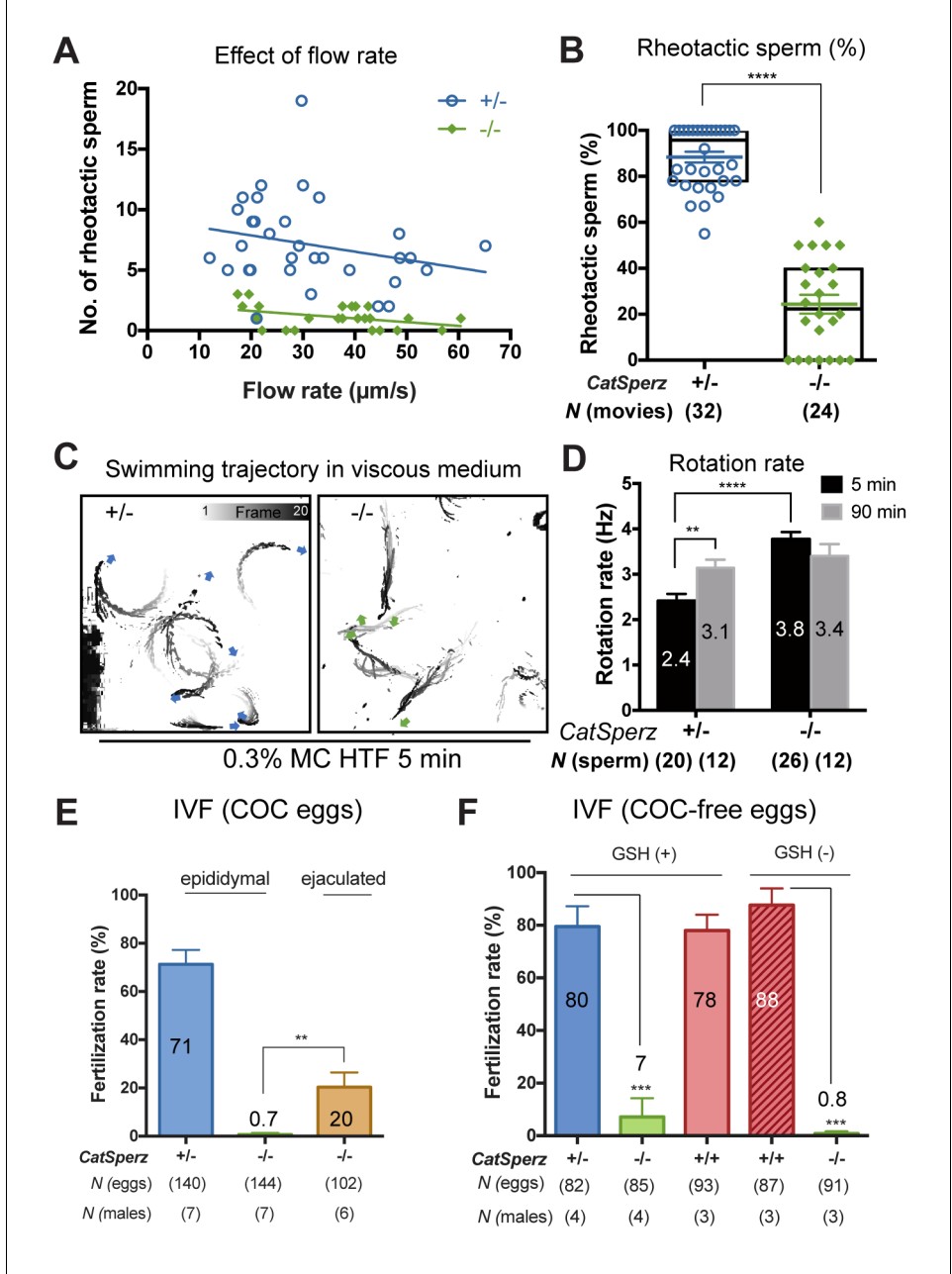

**Figure 6.** *CatSperz*-null sperm rheotax poorly due to low torque. (**A**) In-capillary sperm rheotaxis. Rheotactic ability is reduced in sperm lacking *CatSperz* at all flow rates tested (12–65 µm/s). (**B**) Rheotactic sperm cells are expressed as the % of total motile spermatozoa counted from 9 s-movies (*CatSperz+/-*, n = 32; *CatSperz-/-*, n = 24). Data are expressed in scatter plots; mean ± SEM (colored bars) of *CatSperz+/-* (88 ± 2) and *CatSperz-/-* (24 ± 4) as well as median with interquartile ranges (black boxes) of *CatSperz+/-* (96, IQR 78–100) and *CatSperz-/-* (22.5, IQR 0–40). ****p<0.0001. (**C**) Trajectory of free-swimming sperm in 0.3% methyl cellulose. Movies were taken at 50 fps to compare *CatSperz+/-* (left) and *CatSperz-/-* (right) sperm cells; bottom of glass dish, 37°C, 5 min after incubation in capacitation medium (HTF). Overlays of flagellar traces (20 frames, 2 s movie) are generated by hyperstacking binary images with gray intensity scale; end frame in black. Arrows indicate sperm heads in each trace. (**D**) Sperm rotation rate from *CatSperz+/-* (5 min, 2.4 ± 0.2; 90 min, 3.1 ± 0.2, p=0.0064) and *CatSperz-/-* (5 min, 3.8 ± 0.2; 90 min, 3.4 ± 0.3) males after incubation in HTF. The sperm rotation rate is calculated as previously reported (*Miki and Clapham, 2013*). Data are mean ± SEM. ****p<0.0001. (**E** and **F**) IVF with epididymal and/or ejaculated *CatSperz+/-* and *CatSperz-/-* spermatozoa. 2 cell stage eggs were counted 24 hr after insemination. (**E**) IVF rate with cumulus-intact oocytes from *CatSperz+/-* (epididymal, 71 ± 6) and *CatSperz-/-* (epididymal, 0.7 ± 0.7; ejaculate, 20 ± 6, p=0.0051). (**F**) IVF rate of cumulus-free/ZP-intact eggs with (*CatSperz+/-*, 80 ± 8; *CatSperz-/-*,
*Figure 6 continued on next page*

*Figure 6 continued*

7 ± 7, p=0.0005; wt, 78 ± 6) or without (*CatSperz*+/-, 88 ± 6; *CatSperz*-/-, 0.8 ± 0.8, p=0.0002) glutathione-containing (GSH; 2 mM) media. Data are mean ± SEM. See also *Figure 6—figure supplement 1*.

The following source data and figure supplement are available for figure 6:

**Source data 1.** In-capillary sperm rheotaxis and in vitro fertilization.

**Figure supplement 1.** CatSper-mediated Ca²⁺signaling and development of the flagellar envelope; related to *Figure 6*.

## *CatSperz*-null sperm cells inefficiently penetrate the egg cumulus

We performed *in vitro* fertilization (IVF) to determine how the low rotational torque generated by *CatSperz*-null spermatozoa affects sperm-egg interactions. We found that these spermatozoa cannot fertilize cumulus-intact oocytes (*Figure 6E*), but could dissociate the cumulus cell layers and bind to the ZP (*data not shown*). Cumulus removal did not change the fertilization rate of ZP intact oocytes by *CatSperz*-null spermatozoa. Furthermore, this rate was only marginally enhanced by destabilization of the ZP by 2 mM glutathione (*Figure 6F*) (*Miyata et al., 2015*). This indicates that reduced hyperactivated motility alone does not explain the failure of *CatSperz*-null spermatozoa in IVF. One possibility is that the kinetics of capacitation *in vitro* is different from that *in vivo*, resulting from fluctuations in timing or amplitude of known factors (e.g., $HCO_3$, pH) or from unknown factors present in seminal and/or female fluids. We then compared IVF rates with ejaculated and epididymal spermatozoa of *CatSperz*-null mice. When ejaculated spermatozoa flushed from the uterus of the mated females were used in IVF trials, 20% of oocytes incubated with *z*-null spermatozoa developed into two-cell embryos (*Figure 6E*). This compares to 50% of oocytes incubated with *z*-het ejaculated (*data not shown*), or epididymal sperm. Thus, additional factors may be functionally relevant in *in vivo* fertilization.

## Discussion

### Complex protein composition and conservation of compartmentalization in mammals

Sperm hyperactivation and normal fertility in mammals requires the unique CatSper channel complex. With four distinct pore-forming gene products (CatSper 1–4) and, now, five accessory subunits (β, γ, δ, ε, and ζ), the CatSper channel is the most complex of known ion channels. This may reflect the relatively high evolutionary pressure on spermatozoan evolution (*Swanson and Vacquier, 2002*; *Torgerson et al., 2002*), and various adaptations to different modes of fertilization. Like many gamete-specific proteins and the other CatSper proteins reported so far (*Cai and Clapham, 2008*; *Chung et al., 2011*), mouse and human CatSper ε and ζ show signs of rapid evolutionary change with only 50% and 45% amino acid sequence identity, respectively. In particular, the sequence regions outside TM segments and the pore loop of CatSper proteins are poorly conserved across species, indicating these regions possibly convey species-specific modulation of flagellar motility (*Miller et al.,*

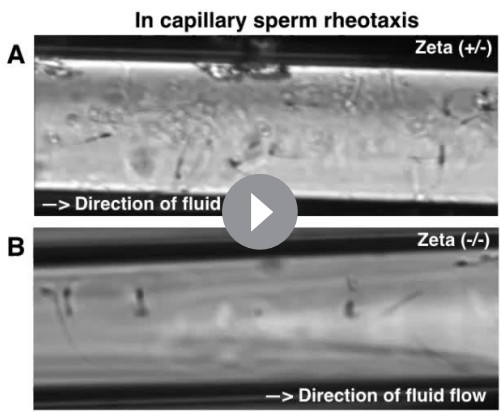

**Video 4.** In-capillary rheotaxis of *CatSperz* +/- and -/- spermatozoa; capacitated, related to *Figure 6*. Capacitated epididymal spermatozoa in HTF for 90 min were loaded into the capillary and transferred to a 37°C chamber; sperm cells swimming against the flow and down were recorded; video rate 33 fps, 9 s movies. (A) *CatSperz*+/- and (B) *CatSperz*-/- spermatozoa.

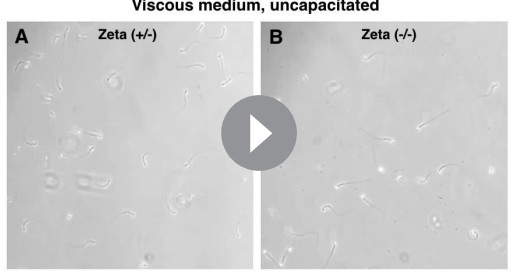

**Viscous medium, uncapacitated**

**Video 5.** Movement of *CatSperz +/-* and *-/-* sperm in viscous medium; uncapacitated, related to **Figure 6**. Uncapacitated spermatozoa were allowed to disperse for 10 min pre-incubation in a 37°C chamber containing HEPES-HTF supplemented with 0.3% methylcellulose; swimming sperm cells were recorded within the next 5 min; video rate 50 fps, 2 s movies. (A) *CatSperz+/-* and (B) *CatSperz-/-* spermatozoa.

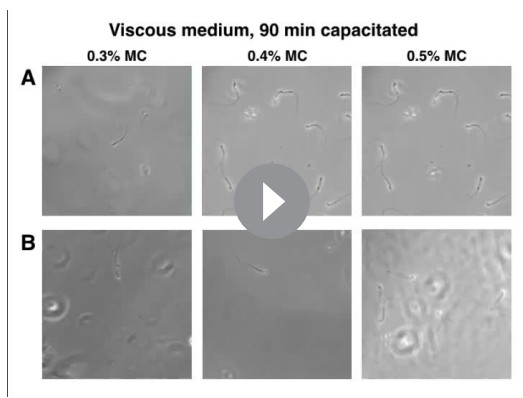

**Viscous medium, 90 min capacitated**

**Video 6.** Movement of *CatSperz +/-* and *-/-* sperm in viscous medium; 90 min capacitated, related to **Figure 6**. Spermatozoa capacitated in HTF were allowed to disperse for 10 min pre-incubation in a 37°C chamber containing HEPES-HTF supplemented with 0.3% (left), 0.4% (middle), or 0.5% (right) methylcellulose; swimming sperm cells were recorded within the next 5 min; video rate 50 fps, 2 s movies. (A) *CatSperz +/-* and (B) *CatSperz-/-* spermatozoa.

2015). This is illustrated by striking differences in progesterone-elicited $I_{CatSper}$ responses in mouse and human (**Lishko et al., 2011**). Here we have shown that CatSper ε and ζ are components of the highly organized CatSper complex, that CatSperζ is required for proper continuity of this complex along the flagellum, and that loss of ζ alters hyperactivation waveforms and reduces fertilizing capacity.

The conservation pattern of the lineage-specific gain and loss of the *CatSpere* gene is identical to those of *b* and *g*, suggesting that they likely belonged to an ancient CatSper channel Ca$^{2+}$ signaling network before the divergence of unikonts and bikonts. Since their protein expression is strictly interdependent, we speculate that *CatSpere*-null mice will have the phenotype of *CatSper1-4*, or *d*-null mice. In contrast, *CatSperz* is conserved only in mammals, suggesting that this protein imparts some adaptation, perhaps as a method enabling rheotaxis in the mammalian female reproductive tract.

Interestingly, although CatSperζ has no putative transmembrane domains, it is localized in the same quadrilateral pattern as other CatSpers, but is not present elsewhere in sperm. An intriguing aspect of our observations is that, unlike *CatSper1-4* and *d*-null mice, which produce complete infertility, *CatSperz*-null males exhibit an incomplete loss of fertility. The CatSper current is reduced in *CatSperz*-null spermatozoa, and may have similar permeation properties (likely dominated by the CatSper1-4 pore subunits), but the effects of *CatSperz* on channel gating remain to be determined in future studies. This is reminiscent of the non-spermatozoan, voltage-gated Ca$_v$ channel auxiliary subunits, which are not required for expression but modulate expression levels and gating (**Catterall et al., 2005**). Most tantalizing is the thinning and disruption of the linear CatSper signaling domains at repeat intervals in the absence of ζ. Further detailed examination via mutagenesis experiments has been stymied by our inability to heterologously express functional CatSper channels. New rapid

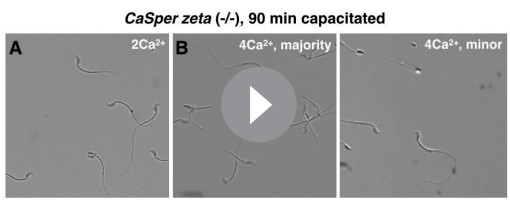

***CaSper zeta* (-/-), 90 min capacitated**

**Video 7.** Motility of tethered *CatSperz-/-* sperm in high external calcium; 90 min capacitated, related to **Figure 6—figure supplement 1**. After 90 min incubation in Ca$^{2+}$-HTF under capacitating conditions, *CatSperz-/-* spermatozoa were tethered to a fibronectin-coated glass bottom dish; sperm motility was recorded within the next 5 min at 37°C; video rate 100 fps (1/2 speed), 1 s movies. (A) 2 Ca$^{2+}$-HTF and (B) 4 Ca$^{2+}$-HTF (in mM).

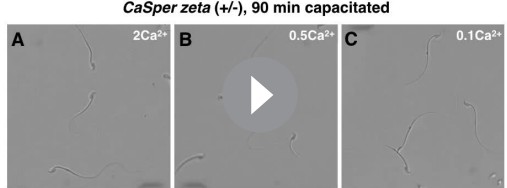

**CaSper zeta (+/-), 90 min capacitated**

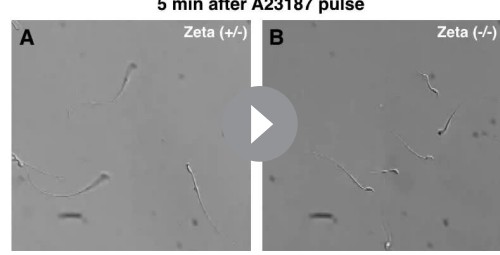

**5 min after A23187 pulse**

**Video 8.** Motility of tethered *CatSperz+/-* sperm in low external calcium; 90 min capacitated, related to *Figure 6—figure supplement 1*. After 90 min incubation in $Ca^{2+}$-HTF under capacitating conditions, *CatSperz+/-* spermatozoa were tethered to a fibronectin-coated glass bottom dish; sperm motility was recorded within the next 5 min at 37°C; video rate 100 fps (1/2 speed), 1 s movies. (A) 2 $Ca^{2+}$-HTF, (B) 0.5 $Ca^{2+}$-HTF and (C) 0.1 $Ca^{2+}$-HTF (in mM). DOI: 10.7554/eLife.23082.028

**Video 9.** Motility of tethered *CatSperz +/-* and *-/-* sperm after A23187 treatment; 5 min after wash, related to *Figure 6—figure supplement 1*. Spermatozoa treated with 20 μM A23187 in H-HTF for 10 min were washed and incubated in HTF under capacitating conditions for 5 min; sperm were tethered to a fibronectin-coated glass bottom dish and the motility was recorded in H-HTF within the next 5 min at 37°C; video rate 100 fps (1/2 speed), 1 s movies. (A) *CatSperz+/-* and (B) *CatSperz-/-* spermatozoa. DOI: 10.7554/eLife.23082.029

genome editing techniques should enable more mice to be generated that will further the study of CatSper trafficking, subunit interactions, and localized signaling pathways.

## Traffic into the linear domains of sperm flagella

Functional $Ca^{2+}$ signaling domains are common adaptations in many biological systems, such as synapses and muscle. They enable specific and fast triggering of downstream events (*Clapham, 2007*). CatSper channels are compartmentalized into a unique multilinear arrangement and form $Ca^{2+}$ signaling nanodomains with other $Ca^{2+}$ signaling molecules along the sperm flagellum (*Chung et al., 2014*). The mechanisms involved in the delivery of the CatSper channels to these specific domains are currently unknown, and we suspect will be as interesting and complex as those in primary and motile cilia (*Sung and Leroux, 2013*). We found that abrogation of *CatSperz* not only retards targeting of the CatSper complex to flagella, but also disrupts continuity of the linear domains, resulting in repeated fragmented domains with ~800 nm periodicity. In order for CatSper domains to form and

function properly, interactions are needed between the CatSper channel complex in the flagellar membrane and the underlying cytoskeletal proteins. One speculation is that CatSperζ might adapt to cytoskeletal structures that traffic, distribute, and enable membrane insertion of CatSper.

The fibrous sheath (FS), a cytoskeletal structure unique to the mammalian sperm flagellum, defines the extent of the tail's principal piece, in which all the CatSper proteins are specifically localized. The FS closely lies under the plasma membrane and its two longitudinal columns are connected by circumferential ribs. Immunogold electron microscopy demonstrated that the CatSper channels are distributed on the end of ribs, where they merge with the column (*Chung et al., 2014*). It seems likely that the timing of occurrence and localization of CatSper $Ca^{2+}$ signaling domains is coordinated with the assembly of FS proteins along the axoneme. The column appears early in

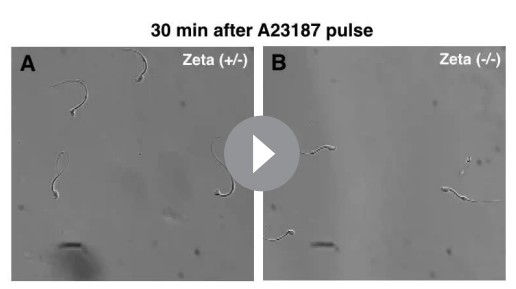

**30 min after A23187 pulse**

**Video 10.** Motility of tethered *CatSperz +/-* and *-/-* sperm after A23187 treatment; 30 min after wash, related to *Figure 6—figure supplement 1*. Spermatozoa treated with 20 μM A23187 in H-HTF for 10 min were washed and incubated in HTF under capacitating conditions for 30 min; sperm were tethered to a fibronectin-coated glass bottom dish and the motility was recorded in H-HTF within the next 5 min at 37°C; video rate 100 fps (1/2 speed), 1 s movies. (A) *CatSperz+/-* and (B) *CatSperz-/-* spermatozoa. DOI: 10.7554/eLife.23082.030

spermiogenesis, forming from the distal tip of the tail along the axoneme, followed by subsequent rib formation in the opposite direction (*Oko, 1998*; *Oko and Clermont, 1989*). Based on scanning electron micrographs (*Danshina et al., 2010*; *Miki et al., 2004*), we find that the distance between ribs is about 800 nm in mouse spermatozoa. Thus, it seems likely that the repeated disruption in the absence of *CatSperz* is related to rib spacing of the FS.

## Ca$^{2+}$ regulation of the flagellar envelope in sperm rheotaxis and egg penetration

Genetic abrogation of *CatSper* disrupts hyperactivated motility as manifested by changes in movement symmetry, amplitude, and rolling (*Carlson et al., 2003*; *Chung et al., 2011*; *Miki and Clapham, 2013*; *Qi et al., 2007*). Here we report that the flagellar envelope is significantly altered in the absence of *CatSperz,* in part due to the inflexibility of the proximal tail. We previously reported that the catalytic subunit of calcineurin, PP2B-A$\gamma$, expresses throughout the tail but localized to the CatSper quadrilateral structures and axoneme (*Chung et al., 2014*). In *CatSper1*-null spermatozoa, PP2B-A$\gamma$ remains localized primarily to the axoneme but disappears from the quadrilateral structures. Recently, a similar but not identical phenotype (inflexible midpiece, reduced hyperactivated motility, and impaired ZP penetration) was reported in testis-specific calcineurin *Ppp3cc*-null and *Ppp3r2*-null spermatozoa (*Miyata et al., 2015*). Note that the principal piece of both *CatSper1*-null and *Ppp3cc*-null spermatozoa are not rigid. The integrity and distribution of CatSper channels in *Ppp3cc*-null spermatozoa remain to be examined and may clarify midpiece/principal piece disparities. In any case, inflexibility in the proximal regions of flagellum results in a flagellar envelope approximated as a rod with a distal propeller. The sperm can rotate faster but the smaller lateral deviation reduces torque. This limits the sperm's ability to orient into the flow, as well as penetrate the cumulus and ZP.

## Physiological modulation of CatSper in sperm function and fertility

Gene-manipulated mice highlight the importance of *in vivo* observations and have reshaped the landscape of fertilization science (*Okabe, 2015*). *In vitro* capacitation and fertilization systems underpin much of the study of sperm motility and fertilization potential. While ejaculated sperm are preferred for fertilization studies in larger animals and humans, epididymal sperm are commonly used in genetically tractable mouse studies. Notably, these sperm are not exposed to accessory sex gland secretions and female fluids. This may explain why *CatSperz*-null spermatozoa are completely infertile in an IVF setting (COCs), but *in vivo* are merely subfertile. Perhaps natural modulators, absent in epididymal sperm IVF studies, partially rescue the fertilizing potential of *CatSperz*-null spermatozoa by activating Ca$^{2+}$ signaling activity.

Interestingly, a transient pulse of Ca$^{2+}$ can greatly reduce the capacitation time required for wt sperm to develop hyperactivated motility (*Tateno et al., 2013*). Moreover, Navarrete et al recently demonstrated that a short exposure to A23187 rescued the defects in motility and fertilizing capability of *CatSper1*-null sperm *in vitro* (*Navarrete et al., 2016*). These independent studies were interpreted to mean that the initial priming by Ca$^{2+}$ influx, perhaps above a certain threshold, is essential for sperm function. However, the linear quadrilateral CatSper complexes are not present in *CatSper1*-null spermatozoa and in *CatSperz*-null spermatozoa are disrupted by gaps. We hypothesize that the linear quadrilateral structure *in vivo* likely maintains, regulates, and distributes CatSper Ca$^{2+}$ signaling during hyperactivated motility. But it is important to point out that alterations in the structure should also result in changes in mechanical properties, movement of the flagellum, distribution of entering calcium, and downstream kinase activity and the motor elements they regulate. This complexity is illustrated *in vivo* sperm swimming trajectories, which are modulated by switching between pro- and anti-hook beating patterns. In the absence of CatSper$\zeta$, anti-hook beating predominates. Pro-hook motions are associated with intact CatSper-mediated Ca$^{2+}$ signaling pathways (*Chang and Suarez, 2011*). Finally, ejaculated sperm display more pro-hook hyperactivation than epididymal sperm (*Li et al., 2015*).

Future areas for investigation are the functional positioning of the remaining accessory subunits of the CatSper channel in assembly and domain organization, the testing of potential modifiers present in accessory sex gland secretions that may activate CatSper channels, and the determination of Ca$^{2+}$ dependent molecules in the axoneme which eventually determine flagellar bending and its

envelope. *CatSperz*-null mice, which are hypomorphic to the null-mutation of other CatSper genes with abrogated hyperactivation, and newly expanding animal models from recent advances in genome editing will serve as a foundation to this end. Advanced imaging techniques with higher time and spatial resolution will be necessary to carry this out. The present results also suggest that alterations of $Ca^{2+}$ current and/or dysregulated downstream $Ca^{2+}$ signaling affecting dynamic structures may be sufficient to compromise sperm function. CatSper's unique composition and central role in hyperactivated motility make it an ideal target for contraception.

## Materials and methods
Details of source and identifier are provided in the Key Resources Table as a supplementary file.

### Animals
*CatSper1* and *d*-null mice were previously described (*Chung et al., 2011*; *Ren et al., 2001*). Lines were backcrossed and maintained on a C57BL/6 background. WT C57BL/6 male, B6D2F1 female (Jackson laboratory, Bar Harbor, ME), and CD1 (Charles River Laboratories, Wilmington, MA) female mice were purchased.

### Generation of CatSperz-deficient mice and genotyping of mutant mice
[1700019N12Rik$^{tm(KOMP)Mbp}$]-targeted ES cells (two clones, 1700019N12Rik_D05 and 1700019N12Rik_C06) were purchased from the UC Davis KOMP repository. The parental ES cell line, JM8A1.N3, was derived from C57BL/6N (agouti) ES cells. Chimeras were born from injection of the C06 ES cells into host embryos. The male chimeras were bred to C57BL/6N females to establish germline transmission and obtain heterozygous animals. Initially, genotype analysis was performed by PCR on isolated genomic DNA (F/R1/R2, F (JJC575): 5'-ATAACCATCCGGGAGGAGAC-3', R1 (YS_zWT-Rev): 5'-GCGATGGTTTGCGTGTTTG-3', R2 (JJC562): 5'- CACAACGGGTTCTTCTGTTAG TCC-3'). From F2 mice, genotyping was done by Transnetyx. The mice used in this study were the offspring of crosses between F1 and/or F2 generations (100% C57BL/6N genetic background). Mice were treated in accordance with guidelines approved by the Boston Children's Hospital and Yale Animal Care and Use Committees (IACUC).

### Mouse sperm preparation and in vitro capacitation
Mouse caudal epididymal sperm were collected by swim-out in HEPES buffered saline (HS) containing (in mM): 135 NaCl, 5 KCl, 2 $CaCl_2$, 1 $MgSO_4$, 20 HEPES, 5 glucose, 10 lactic acid, 1 Na pyruvate, pH 7.4 (with NaOH) (*Chung et al., 2011*). To induce capacitation *in vitro*, sperm cells were incubated ($2 \times 10^6$ cells ml$^{-1}$) in human tubular fluid (HTF) media (in mM): 102 NaCl, 4.7 KCl, 2 $CaCl_2$, 0.2 $MgCl_2$, 0.37 $KH_2PO_4$, 2.78 glucose, 18.3 lactic acid, 0.33 Na pyruvate, 25 $HCO_3^-$ and 4 mg ml$^{-1}$ BSA) (Millipore) for 90 min at 37°C (5% $CO_2$).

#### Capacitation in varying external [$Ca^{2+}$]
To test the effect of external [$Ca^{2+}$] on the development of hyperactivated motility, 2 mM $CaCl_2$ in standard HTF was replaced with 4 mM, 0.5 mM, and 0.1 mM $CaCl_2$ and sperm cells were incubated for 90 min under capacitating conditions (37°C, 5% $CO_2$).

#### Motility rescue by $Ca^{2+}$ ionophore, A23187 treatment
$Ca^{2+}$ transient-inducible hyperactivated motility was tested by treating sperm with A23187 as described (*Navarrete et al., 2016*; *Tateno et al., 2013*) with slight modification. In short, caudal epididymal mouse sperm were collected by swim-out in HEPES-HTF medium (H-HTF: 92 mM NaCl, 2 mM $CaCl_2$, 4.7 mM KCl, 0.2 mM $MgCl_2$, 0.37 mM $KH_2PO_4$, 25 mM $NaHCO_3$, 18.3 mM Na lactate, 2.78 mM glucose, 0.33 mM Na pyruvate, 0.4% [w/v] bovine serum albumin [BSA], and 10 mM HEPES [pH 7.4]), allowing motile sperm to disperse for 10 min at 37°C. Sperm concentration was ~$4 \times 10^7$ cells/mL. An aliquot (150 µL) of the sperm suspension was exposed to 20 µM A23187 in H-HTF. After 10 min, sperm were washed by two centrifugation at 550 G for 5 min and 300 G for 5 min, resuspended in standard HTF, and incubated at $2 \times 10^6$ cells/ml under capacitating conditions (37°C, 5%

$CO_2$). At 5 min and 30 min post-incubation, spermatozoa were tethered to fibronectin-coated coverslips and recorded in H-HTF at 37°C as described in *Flagellar Waveform Analysis*.

## Human sperm preparation

All experiments using human samples were approved by the Committee of Clinical Investigation, Boston Children's Hospital CCI/IRB (IRB-P00000538). Human semen samples were obtained from fertile donors. Human spermatozoa were collected by the swim-up method with the use of modified human tubal fluid medium (HTF).

## Cell line origin and authentication

HEK293T cells were purchased from ATCC. In this study, they were used to overexpress recombinant human CatSperζ in order to test antibodies. The cells were tested negative for mycoplasma and validated as of human origin. The identity was authenticated by confirming their negative expression of testis-specific genes including *CatSper*. The cell line was cultured in DMEM/F12 containing 10% FBS.

## Antibodies and reagents

Rabbit polyclonal CatSper1, CatSper4, β, and δ antibodies were previously described (*Chung et al., 2011*; *Ren et al., 2001*). To produce antibodies to new CatSper subunits, peptides were synthesized and conjugated to KLH carrier protein (Open Biosystems, Lafayette, CO) as follows: mouse CatSperε, 968–985 (αm-ε968: RQFIIEPLHKRPAKQKKN); mouse CatSperζ, 174–195 (αm-ε174: G YIEGIRKRRNKRLYFLDQ); human CatSperε, 31–50 (αh-ε31: RIFSTRSTIKLEYEGTLFTE); and human CatSperζ, 11–29 (αh-ζ11: KSSDRQGSDEESVHSDTRD). Antisera were affinity purified on the immobilized resin of the corresponding peptide (Amino Link Plus or Sulfo Link Plus) (Pierce, Waltham, MA). Anti-phosphotyrosine (clone 4G10), anti-Flag (clone M2), anti-calmodulin (05-173) and anti-acetylated tubulin (T7451) antibodies were from EMD Millipore (Germany). All chemical compounds were from Sigma-Aldrich (St. Louis, MO) unless indicated.

## Genomic database search

Annotated orthologs in the NCBI gene database (http://www.ncbi.nlm.nih.gov/gene/) and/or homologous amino acid sequences of reported protein databases were screened in 17 eukaryotes for the presence of genes for CatSper auxiliary subunits. Non-annotated orthologs in the NCBI gene database were identified by comparing sequences of the annotated orthologs to those in the protein database of species by Phmmer implemented on HMMER 3.1 (default option, http://hmmer.org/). The longest amino acid sequences among all the isoforms of the orthologs annotated in each species and protein sequence databases from 15 eukaryotes, except human and mouse, were downloaded from the NCBI Genome database (http://www.ncbi.nlm.nih.gov/genome; *Tinamus guttatus*, GCA000705375.2; *Anolis carolinensis,* GCA000090745.2; *Salmo salar*, GCA000233375.4; *Callorhinchus milii*, GCA000165045.2; *Branchiostoma floridae*, GCA000003815.1; *Caenorhabditis elegans*, GCA000002985.3; *Crassostrea gigas*, GCA000297895.1; *Exaiptasia pallida*, GCA001417965.1; *Trichoplax adhaerens*; GCA000150275.1, *Salpingoeca rosetta*, GCA_000188695.1), Ensembl genome browser (http://ensembl.org; *Strongylocentrotus purpuratus*, GCA000002235.2; *Drosophila melanogaster*, GCA000001215.4; *Thecamonas trahens,* GCA000142905.1), and JGI genome portal (http://genome.jgi.doe.gov; *Allomyces macrogynus; Aurantiochytrium limacinum*). Aligned phmmer hits of expected values <$10^{-10}$ were considered as candidate orthologs of the corresponding CatSper subunits in each species.

## Multiple tissue RT–PCR

PCR was performed according to standard protocols using a commercial multiple panel cDNA template (MTC), Clontech). PCR primers amplified *Gm7068* (forward: 5′-CTATGGCTCAAGTGTAA TGACC-3′, reverse: 5′-GCTCTTATTGAATCCTCGAACC-3′), *Tex40* (forward: 5′-GAAACAGGA TTCGCAAGTACAG-3′, reverse: 5′-TCGTGGACCTATATGTGATGAG-3′) using mouse *GAPDH* (forward: 5′-TGAAGGTCGGTGTGAACGGATTTGGC-3′, 5′-ATGTAGGCCATGAGGTCCACCAC-3′) as a control.

## Molecular cloning

The initial mouse *Tex40* cDNA sequence (NM_001039494) was identified from database searches using novel peptide sequences from MS. The full-length human *Tex40* cDNAs was obtained by PCR with primers (forward: 5′-GGGCAGAACCATGGAGGAAA-3′, reverse: 5′-AGGACTCAAATTCCAC TCGGATG-3′) using the human testis cDNA library (Clontech). Sequencing the TOPO-cloned PCR products into pCR4-TA (Invitrogen) confirmed the full-length human *Tex40* ORF, which was subcloned into pCMV-Tag2A (Stratagene) to express recombinant N-terminal Flag-tagged human CatSperζ in mammalian cells. Mouse *Gm7068* was identified by homologous amino acid sequence to C-terminal *CatSperd* (Tmem146). There are six transcript variants (*Almers et al., 1984*); XM_006497083, 2; XM_006497084, 3; XM_017314031, 5: XM_006497085, 6; XM_017314033, and 8; XM_006497087. Variants 1, 3, 5, 6, and 8 are predicted to encode polypeptides with the same C-terminal sequence that can be detected by anti-mε−968. Among them, the predicted polypeptides from longer splicing variant 1 (isoform X1; XP_006497147, 985 aa) and variant 3 (isoform X3; XP_017169520, 914 aa) are consistent with the apparent molecular weight of the band observed in testes microsomes (*Figure 4C* and *Figure 1—figure supplement 2D*). The predicted polypeptides from shorter variant 5 (isoform 4; XP_006497148, 805 aa) and variant 6 (isoform 5; XP_017169522, 770 aa) are consistent with that of the band detected in CatSper1-IP from testis and total sperm lysate (*Figures 1D,E and* and *4A*). It is likely that mouse *Gm7068* expresses at least four potential splice variants that can encode protein isoforms and/or undergo cleavage during spermatogenesis.

## RNA *in situ* hybridization

In situ hybridization experiments were carried out with an RNAscope (Advanced Cell Diagnostics, Newark, CA). Testes from three month old wild-type mice were fixed in 10% (vol/vol) neutral-buffered formalin at room temperature for 24 hr, dehydrated, and embedded in paraffin. Paraffin sections (10 μm thick) were processed according to the manufacturer's instructions for in situ detection in the Rodent Histopathology Core Facility at Harvard Medical School. Sequences of the probes used in this study are: *Gm7068* (XM_982472.3, 645–1072) and *Tex40* (NM_001039494.2, 41–456). After the DAB (3,3, -diaminobenzidine) reaction, slides were counterstained using hematoxylin.

## mRNA preparation and Real-time PCR

Real-time PCR was carried out with first strand cDNAs (iScript cDNA Synthesis) (Bio-Rad, Hercules, CA) synthesized from 2 μg total mouse testis RNA using the SYBR Green (iTaq Universal SYBR Green Supermix) (Bio-Rad; CFX96). Quantitative analysis by the dd$Ct$ method employed c-Jun as an amplification control. Three independent sets of experiments were performed to calculate fold changes ($2^{-ddCt}$) of *CatSpers* mRNA. The primers used for qRT-PCR were: *CatSper1* (forward: 5′-CTGCCTCTTCCTCTTCTCTG-3′, reverse: 5′-TGTCTATGTAGATGAGGGACCA-3′), *CatSperb* (forward: 5′-CCTTA TTGACCAAGAAACAGAC-3′, reverse: 5′-TGAAACCCATATTTGACTGCC-3′), *CatSperg* (forward: 5′-TGAGCAATAGAGGTGTAGAC-3′, reverse: 5′-CAGGA TGTAGAAGACAAC-CAG-3′), *CatSperd* (forward: 5′-GCTGACATTTCTGTGTATCTAGG-3′, reverse: 5′-CTGATATACC TTCCAATTTACGCC-3′), *CatSpere* (forward: 5′-GTCTCATGCTTCTTCAGTTCC-3′, reverse: 5′-CAGAAGTTCCTTGTCCATCAC-3′), *CatSperz* (forward: 5′-GAGACCTCCTTAGCATCGTC-3′, reverse: 5′-TCGTGGACCTATATGTGATGAG-3′ and *c-Jun* (forward: 5′-CTCCAGACGGCAGTGCTT-3′, reverse: 5′-GAGTGCTAGCGGAGTCTTAACC-3′).

## Preparation of mouse testis microsome

Testes (200 mg, normally two testicles) from 8- to 12-wk-old male mice were homogenized on ice using a Dounce homogenizer in 2 mL 0.32 M sucrose solution with protease inhibitor cocktails (Roche). The tissue suspension was centrifuged at 300 g for 10 min at 4°C and the supernatant was then transferred to an ultra-speed centrifuge tube. The microsome faction was isolated by centrifuging the tube at 105,000 g for 60 min.

## Protein preparation, immunoprecipitation, and western blotting

Mouse sperm total protein was prepared as described before (*Chung et al., 2011*; *2014*). For total protein from human spermatozoa, purified swim-up sperm were then lysed (0.1% SDS, 0.5% sodium

deoxycholate, 1 mM DTT, 1 mM EDTA in PBS with protease inhibitors) followed by sonication for 5 min and centrifuged at 15,000 g for 10 min. The supernatants were further denatured by adding DTT to 10 mM and heated at 75°C for 10 min before SDS-PAGE. For immunoprecipitation, the testis microsome pellet was resuspended in 10 mL 1% Triton X-100 in PBS with protease inhibitors (Roche). The suspension was rocked at 4°C for 1 hr and then centrifuged at 15,000 g for 30 min. 1.5 mL of the solubilized testis microsome were mixed with 1–2 µg antibody and 25 µL Protein A/G-bead slurry (Santa Cruz Biotechnology) at 4°C overnight. The IP products were finally eluted in 50 µL LDS loading buffer containing 50 mM DTT. Antibodies used for Western blotting were rabbit anti-mouse CatSperε (αm-ε968; 1.6 µg/mL), mouse CatSperζ (αm-ε174; 2.7 µg/mL), human CatSperε (αh-ε31; 2.7 µg/mL), human CatSperζ (αh-ζ11: 1 µg/mL). Monoclonal anti-phosphotyrosine (clone 4G10; 1 µg/mL), anti-Flag (clone M2; 1 µg/mL), anti-calmodulin (05–173, 1 µg/mL), and anti-acetylated tubulin (T7451,1: 20,000). Secondary antibodies were anti-rabbit IgG-HRP (1:10,000) and anti-mouse IgG-HRP (1: 10,000) from Jackson ImmunoResearch (West Grove, PA).

## Sperm immunocytochemistry

Caudal epididymal mouse sperm cells attached to glass coverslips were fixed in 4% paraformaldehyde (PFA) in PBS, permeabilized with 0.1% TrixonX-100 for 10 min. Human sperm cells from swim-up purification were fixed 4% PFA in PBS for 10 min followed by 100% MeOH. Fixed human sperm cells were permeabilized in 0.1% saponin for 10 min. Permeabilized sperm cells were washed in PBS and blocked with 10% goat serum for 1 hr. Mouse samples were stained overnight with primary antibody against CatSper1 (10 µg ml$^{-1}$) and CatSperζ (mζ174, 20 µg ml$^{-1}$) as were human samples with primary antibodies against CatSperε (hε31, 20 µg ml$^{-1}$) and CatSperζ (hζ11, 10 µg ml$^{-1}$), in 10% goat serum in PBS, 4°C. After PBS wash, goat-anti-rabbit Alexa488 conjugate (Invitrogen) served as the secondary antibody. Images were acquired on laser scanning confocal microscopes (Olympus Fluoview 1000; *Figure 1G and H*, *Figure 1—figure supplement 2G*, and *Figure 2—figure supplement 1E* and Leica TCS SP8; deconvolved image in *Figure 5A*).

## Super-resolution imaging

### 3D STED imaging

For analysis of CatSper nanodomain organization, CatSper1 images were acquired with Leica TCS SP8 gated stimulated emission and depletion (STED 3×) microscopy using an HCX PL APO 100×/1.40 oil objective lens (Leica Microsystems, Germany). Samples were prepared as described in *Sperm Immunocytochemistry* with slight modifications. After incubation with primary antibody, cells were washed with PBS and incubated with goat anti-rabbit IgG coupled to Alexa Fluor 546 (Invitrogen, 1:100) for 1 hr at room temperature. Coverslips were mounted with Prolong Gold (Invitrogen) and cured for 24 hr before image acquisition. Within each experiment, identical settings for laser power, STED power, and gating were used to acquire images. The wavelength of the STED depletion laser was 660 nm and was adjusted to 50% of power. z-stacks of 17 optical sections with a step size of 0.1 µm were deconvolved using Huygens Software.

### 3D STORM imaging

3D STORM experiments were performed as previously described (*Chung et al., 2014*). Imaging buffer was prepared in 60% (wt/wt) sucrose solution, increasing imaging depth to 1 µm (*Chung et al., 2014*). Imaging buffer was supplemented with 100 mM mercaptoethylamine (pH 8.5) as a switching agent as well as an $O_2$ scavenger (5% glucose (wt/vol), 0.5 mg/ml glucose oxidase, and 40 mg/ml catalase) to reduce the rate of photobleaching. The sample was illuminated at 657 nm for imaging the photoswitchable reporter molecules (Alexa 647), and 405 or 532 nm for facilitating the activation of Alexa 647 from the dark state. For 3D localization, a cylindrical lens (focal length = 1 m) was inserted into the detection path to enable determination of z positions from the ellipticities of the molecular images and the x and y positions from the centroid positions (*Huang et al., 2008*). Image analysis and rendering was performed and angular profiles were constructed as previously described (*Chung et al., 2014*).

## Fourier transform and autocorrelation

Fourier transform and autocorrelation analyses of 3D STORM images were performed as previously described (*Xu et al., 2013*; *Zhong et al., 2014*). A Fourier transform of the 1D projection localization distribution yielded a main peak that corresponds to a spatial period of ~800 nm for the *CatSperz-/-* spermatozoa. The autocorrelation curve for the *CatSperz-/-* spermatozoa showed a periodic modulation with the first peak at ~850 nm.

## Sperm migration assay and *in vivo* fertilization

For timed coitus, females were introduced to single-caged *CatSperz*-het or -null males for 1 hr and checked for the presence of a vaginal plug. To examine sperm migration to the fertilization site *in vivo*, ampullae were removed from the mated females at 8 hr after coitus and COCs were released. A series of *z*-stacked images (2 μm step size) of the COCs was taken and number of sperm within each COC was recorded according to the presence of a sperm head. To calculate *in vivo* fertilization rate, eggs were gently flushed from oviducts and ampullae from the mated females at 20 hr and 27–30 hr after coitus. The total number of eggs and the number of 2 cell eggs were counted.

## Fertility test and *in vitro* fertilization

Two females were caged with each male for three months to track pregnancy and litter production. For IVF assays, oocytes were recovered from superovulated 5–6-week-old B6D2F1 female mice 13 hr after injection of 5 U human chorionic gonadotropin. For standard IVF, sperm were collected from the *cauda* epididymis. For ejaculate IVF, sperm were retrieved from the uterus of a 1 hr window-timed coitus. Both epididymal and ejaculated sperm were capacitated *in vitro* at 37°C for 1 hr, and coincubated with eggs at ~$10^5$ sperm/mL. After about 4.5 hr, unbound sperm were washed away. After 24 hr incubation the embryos were observed under light microscopy (Olympus IX-70) to check for development of the two-cell stage.

## Flagellar waveform analysis

Spermatozoa from the dissected *cauda* epididymis (swim up method) were collected in HEPES buffered saline (HS) media. Spermatozoa were plated on 35 mm fibronectin-coated coverslips for 15 min (22°C); unattached sperm were removed by the gentle pipette wash (time 0) and basal motility recorded. Activated motility was recorded within the first 10 min after adding pre-warmed human tubal fluid (HTF)-capacitating medium (Millipore). To induce hyperactivation, attached sperm cells were incubated in HTF media for 90 min at 37°C (5% $CO_2$). All subsequent images were recorded at 37°C. The flagellar waveform was analyzed by stop-motion digital imaging collected at 200 fps (HC Image software, Hamamatsu Photonics or Zen Blue, Zeiss; 2 s movies). Overlay of flagellar traces from two complete flagellar beats were generated by hyperstacking binary images using open-source FIJI software (*Schindelin et al., 2012*) and time coded in color.

## Sperm motility analysis

Cauda epididymal spermatozoa were suspended and incubated in non-capacitating M2 medium (Specialty Media, Millipore) or in HTF medium for capacitation. Sperm motility was then measured using the IVOS sperm analysis system (Hamilton Thorne Biosciences, Beverly, MA) in an 80 μm (depth) chamber to obtain various parameters (*Figure 2—figure supplement 2D*). Sperm motility was also analyzed with an Olympus IX-70 microscope equipped with a high-speed sCMOS camera (Orca-Flash4.0) and a 10x objective. 1–2 × $10^5$ mouse sperm before and after capacitation were added to the 37°C chamber (Delta T culture dish controller; Bioptechs) containing 1 ml HEPES-HTF medium (H-HTF: 92 mM NaCl, 2 mM $CaCl_2$, 4.7 mM KCl, 0.2 mM $MgCl_2$, 0.37 mM $KH_2PO_4$, 25 mM $NaHCO_3$, 18.3 mM Na lactate, 2.78 mM glucose, 0.33 mM Na pyruvate, 0.4% [w/v] bovine serum albumin [BSA], and 10 mM HEPES [pH 7.4]). In some experiments, the medium was supplemented with methylcellulose (MC) (M0512, 4000 cP in 2% solution; Sigma) at 0.3%, 0.4%, or 0.5% (w/v). Sperm swimming 3–5 mm from the rim were recorded after a 10 min preincubation period that allowed spontaneous dissociation of sperm clumps. To inhibit convective flow, 1 ml of medium was overlaid by 1 ml of mineral oil and covered by a heated glass lid (Bioptechs). Sperm motility at 37°C was videotaped at 100 fps. Images (HC Image software, Hamamatsu Photonics) were analyzed for swimming trajectory from a 1 s playback movie at 1/5 speed, by head tracing via Computer Assisted

Sperm Analysis (CASA; http://rsbweb.nih.gov/ij/plugins/casa.html). To track swimming trajectory in viscous medium, the sperm motility was videotaped at 50 fps. The images were analyzed using Fiji software (Schindelin et al., 2012) by assembling overlays of the flagellar traces generated by hyper-stacking binary images of 20 frames of 2 s movies coded in a gray intensity scale.

### In-capillary sperm rheotaxis

Mouse sperm incubated in HTF medium for 90 min at $2 \times 10^6$/ml yielded capacitated sperm. Capacitated sperm were transferred and concentrated for capillary loading by centrifugation at 900 g for 3 min. The loose sperm pellet at the bottom of the microcentrifuge tube was resuspended in HEPES-HTF at $4 \times 10^6$/ml, and loaded into the capillary by suction via an air-pressure microinjector (IM-5B; Narishige; 22 C, ~200 μm/s). While applying gentle positive pressure, the sperm in the tip of the capillary were moved out of the sperm drop. The tip of the capillary is transferred to a 37°C chamber (Delta T culture dish controller; Bioptechs) and placed into a 50 μl drop of HEPES-HTF medium covered with mineral oil. Negative pressure was applied slowly and sperm cells swimming against the flow and down to the H-HTF drop was video-recorded at 33 fps.

### Electrophysiological recording of mouse spermatozoa

Whole-cell recording of *corpus* epididymal spermatozoa from 3–5 month-old *CatSperz+/-* or *Cat-Sperz-/-* mice was performed blind as to genotype (Kirichok et al., 2006; Navarro et al., 2011). HS was the bath medium. The standard pipette solution was (mM): 120 Cs- Methanesulfonate (Cs-MeSO$_4$), 5 CsCl, 5 Cs-BAPTA, 10 HEPES and 10 MES, pH 7.2 with H-MeSO$_3$. To record $I_{ATP}$, we used a low Cl$^-$ bath solution (to reduce background Cl$^-$ conductance) in the following (mM): 150 Na-methanesulfonate (Na-MeSO$_3$), 2 CaCl$_2$, 10 Na-HEPES, and 10 MES (pH 7.4 or 6.0). To measure $I_{CatSper}$, we used divalent-free (DVF) solution, in mM: 150 Na-MeSO$_3$, 2 Na$_3$HEDTA [(hydroxyethyl)ethylenediaminetriacetic acid], 2 EGTA, and 20 HEPES (pH 7.4) with NaOH. Solutions were applied to sperm cells (lifted from the coverslips) initially by bath perfusion. After break-in, the access resistance was 25–80 MΩ. All experiments were performed at 22–24°C. The whole-cell currents were recorded using an Axopatch 200B amplifier (Molecular Devices, Sunnyvale, CA), acquired with Clampex 9 (pClamp9 Software; Molecular Devices), and analyzed with Origin software (OriginLab). Signals were low-pass filtered at 2 kHz and sampled at 10 kHz. Data are given as mean ± SD.

### Quantification and statistical analysis

All the experiments are repeated at least three times. Sample size and number of replicates are described in each figure and the figure legends. Statistical analyses were performed using Student's t-test unless indicated; e.g. F-test in one-way ANOVA. Differences were considered significant at *p<0.05, **p<0.01, ***p<0.001, and ****p<0.0001. When ****p<0.0001, actual *P* value is not indicated.

## Acknowledgements

We thank M. Sanguinetti for testing CatSper expression in *Xenopus* oocytes for $I_{CatSper}$ detection, P. DeCaen for help with sperm electrophysiology, and L. Ded for calculating rib length from the literature and critical reading of the manuscript-in-progress. This work was supported by Goodman-Gilman Yale Scholar Award 2015–08 (J-JC), HHMI (XZ), and HHMI (DEC).

## Additional information

### Funding

| Funder | Grant reference number | Author |
|--------|------------------------|--------|
| Howard Hughes Medical Institute | | Xiaowei Zhuang<br>David E Clapham |
| Yale School of Medicine | Goodman-Gilman Yale Scholar Award 2015-08 | Jean-Ju Chung |

The funders had no role in study design, data collection and interpretation, or the decision to submit the work for publication.

## Author contributions
J-JC, Conceptualization, Resources, Data curation, Formal analysis, Supervision, Funding acquisition, Validation, Investigation, Visualization, Methodology, Writing—original draft, Project administration, Writing—review and editing, Conceived and supervised the project, Designed, performed and analyzed experiments: mouse genetics, gene expression studies, protein biochemistry including immunoblotting, confocal and STED imaging, in vitro and in vivo fertilization assays, sperm motility and waveform analyses, and rheotactic measurement of sperm, Contributed to all of the STORM imaging experiments, Assembled figures and wrote the manuscript with the input from the co-authors; KM, Data curation, Formal analysis, Performed and analyzed electrophysiological recording, and contributed to rheotactic measurement of sperm; DK, S-HS, Data curation, Formal analysis, Visualization, Performed STORM imaging, image analysis, and rendering; HFS, Data curation, Formal analysis, Contributed to immunoblotting, in vitro and in vivo fertilization assays, rheotactic measurement of sperm, and assembled figures together with J-JC; JYH, Data curation, Formal analysis, Did comparative genomic analysis and contributed to sperm motility and waveform analyses; XC, Data curation, Did comparative genomic analysis; YI, Data curation, Contributed to mouse genetics and in vitro fertilization assay; XZ, Supervision, Funding acquisition, Writing—review and editing, Advised DK and SHS on STORM experiments; DEC, Supervision, Funding acquisition, Writing—original draft, Writing—review and editing, Supervised the project and wrote the manuscript with the input from the co-authors

## Author ORCIDs
Jean-Ju Chung, http://orcid.org/0000-0001-8018-1355
Huanan F Shi, http://orcid.org/0000-0003-3710-5917
Xinjiang Cai, http://orcid.org/0000-0001-8933-7133
Xiaowei Zhuang, http://orcid.org/0000-0002-6034-7853
David E Clapham, http://orcid.org/0000-0002-4459-9428

## Ethics
Animal experimentation: This study was performed in strict accordance with the recommendations in the Guide for the Care and Use of Laboratory Animals of the National Institutes of Health. All the mice were treated in accordance with guidelines approved by the Boston Children's Hospital (13-01-2341R) and Yale (2015-20079) Animal Care and Use Committees (IACUC).

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
