## [Decision Letter]

Thank you for submitting your article "CatSperζ Regulates the Structural Continuity of Sperm Ca^2+^ Signaling Domains and is Required for Normal Fertility" for consideration by *eLife*. Your article has been reviewed by three peer reviewers, one of whom, Kenton Swartz, is a member of our Board of Reviewing Editors, and the evaluation has been overseen by and Richard Aldrich as the Senior Editor. The following individual involved in review of your submission has agreed to reveal his identity: Christopher J Lingle (Reviewer #3).

The reviewers have discussed the reviews with one another and the Reviewing Editor has drafted this decision to help you prepare a revised submission.

Summary:

In this very interesting manuscript, Chung et al. report the discovery and the function of two novel subunits of the CatSper complex, CatSperε and CatSperζ. The CatSper channel has been known for >10 years now to form the most important Ca^2+^ channel in sperm tail and is absolutely required for male fertility. However, the complex has been resistant to detailed functional studies by biophysicists because it somehow doesn't form "functional" channel in heterologous expression system. One explanation is that we don't have all the subunits in the complex yet despite the 4 pore-forming (CatSper1-4) and 3 "auxiliary" (CatSperβ, γ, δ) reported thus far. The discovery of another two subunits described in this paper is a major step forward. Intriguingly, CatSperζ doesn't have a transmembrane-spanning domain and is yet membrane-anchored (perhaps in a fashion similar to the β subunits of CaVs). The authors show that during evolution, CatSperζ was "added" to the complex in mammals and it presumably reflects an adaptation to the changes of environment of the site of fertilization in the female reproductive tract. The authors convincingly demonstrate that the new proteins are associated with CatSper (using biochemical methods), colocalized within the CatSper nanodomains (with STROM and conventional immuno-staining) and, for CatSperζ, are required for the continuity of the CatSper structural stripes along the sperm tail. Finally, the authors demonstrate that CatSperζ is required for normal sperm hyperactivated motility, egg penetration and male fertility (sub-fertile in the KO).

The findings in the paper are novel, interesting and convincing. The studies are remarkably comprehensive, and include data from the identification of new subunits to the biochemical characterization of interaction and whole-organism function. The following are suggestions the authors should carefully consider in preparing a revised manuscript.

Suggested revisions:

1) "Auxillary" subunits in other channels (e.g. CaV β and NaV β) usually affect the channel's voltage dependence of activation/inactivation kinetics. In Figure 3, the authors present the averaged amplitudes of *I_CatSper_* in the KO and WT. It would be nice to have the averaged G-V relationships reconstructed from data in Figure 3. In addition, it's useful to show the traces recorded with square wave stimulation, in addition to the ones with ramp protocols, to check whether there is any inactivation in the KO sperm. Finally, does the channel without CatSperζ have the same pH-dependence?

2) There was a concern that the authors have not carefully distinguished to what extent the sperm motility deficits in the zeta KO arise from reductions in *I_CatSper)_*vs. structural changes in localization and association of the CatSper complex. In the minor comment 2 below, there is a suggestion for potential tests of this issue. At a minimum, some cautionary remarks on this potential distinction are required.

---

## [Author Response]

Suggested revisions:

1) "Auxillary" subunits in other channels (e.g. CaV β and NaV β) usually affect the channel's voltage dependence of activation/inactivation kinetics. In Figure 3, the authors present the averaged amplitudes of I_CatSper_ in the KO and WT. It would be nice to have the averaged G-V relationships reconstructed from data in Figure 3. In addition, it's useful to show the traces recorded with square wave stimulation, in addition to the ones with ramp protocols, to check whether there is any inactivation in the KO sperm. Finally, does the channel without CatSperζ have the same pH-dependence?

These are excellent suggestions. Unfortunately, the person who did those recordings, Kiyoshi Miki, had to return to his family in Japan. After going through all his data, we don’t have enough step protocol data to provide adequate G-V curves. With both of us, the lead authors, moving and starting new labs, we will have to leave this for future work. In any case, it will take many months, beyond the *eLife* limit, due to the slow pace of (our) sperm recording.

2) There was a concern that the authors have not carefully distinguished to what extent the sperm motility deficits in the zeta KO arise from reductions in I(CatSper) vs. structural changes in localization and association of the CatSper complex. In the minor comment 2 below, there is a suggestion for potential tests of this issue. At a minimum, some cautionary remarks on this potential distinction are required.

We agree. We revised the text to clarify the reviewers’ point in subsection “Reduced CatSper Current in CatSperζ-null Spermatozoa.”, second paragraph and added a new section in the Results section “Compromised Ca2+ Signaling Alters the Sperm’s 3D Flagellar Envelope and Movement “. We now also reference work done by Navarette and colleagues—their work recently showed that motility and fertility could be rescued in CatSper KO spermatozoa via A23187 pulsed treatment to induce calcium transients (Navarrette et al., 2016). This directly addresses the question as it demonstrates that the Ca^2+^ transients resulting from A23187 pulsed treatment are sufficient to induce sperm hyperactivation in vitro, bypassing not only the pore but also the spatial arrangement of the CatSper channel complex. We also performed additional motility experiments in varying concentrations of extracellular Ca^2+^ or with A23187 pulses to determine whether these conditions rescue the observed phenotype of CatSperζ-KO sperm. These results are now added to the main text in the Results and Discussion sections.